# LAION-C: An Out-of-Distribution Benchmark for Web-Scale Vision Models

Fanfei Li [1]   Thomas Klein [1]   Wieland Brendel [1 2 3]   Robert Geirhos [‡ 4]   Roland S. Zimmermann [‡ 4]

## Abstract

Out-of-distribution (OOD) robustness is a desired property of computer vision models. Improving model robustness requires high-quality signals from robustness benchmarks to quantify progress. While various benchmark datasets such as ImageNet-C were proposed in the ImageNet era, most ImageNet-C corruption types are no longer OOD relative to today's large, web-scraped datasets, which already contain common corruptions such as blur or JPEG compression artifacts. Consequently, these benchmarks are no longer well-suited for evaluating OOD robustness in the era of web-scale datasets. Indeed, recent models show saturating scores on ImageNet-era OOD benchmarks, indicating that it is unclear whether models trained on web-scale datasets truly become better at OOD generalization or whether they have simply been exposed to the test distortions during training. To address this, we introduce LAION-C as a benchmark alternative for ImageNet-C. LAION-C consists of six novel distortion types specifically designed to be OOD, even for web-scale datasets such as LAION. In a comprehensive evaluation of state-of-the-art models, we find that the LAION-C dataset poses significant challenges to contemporary models, including MLLMs such as Gemini and GPT-4o. We additionally conducted a psychophysical experiment to evaluate the difficulty of our corruptions for human observers, enabling a comparison of models to lab-quality human robustness data. We observe a paradigm shift in OOD generalization: from humans outperforming models, to the best models now matching or outperforming the best human observers.

## 1. Introduction

In recent years, large-scale vision models such as vision transformers (Dosovitskiy et al., 2021) and ConvNeXt (Liu et al., 2022), trained on expansive web-scale datasets like LAION (Schuhmann et al., 2022), have pushed the boundaries of performance on standard benchmarks. However, the continued advancement and reliable evaluation of these models depends on the availability of datasets that effectively challenge model robustness and generalization capabilities.

In the era of training models on curated datasets like ImageNet (Russakovsky et al., 2015), creating OOD-benchmarks was relatively straight-forward: By introducing visual corruptions that were absent from the training set, such as blur and noise, researchers could evaluate the robustness of their models in a controlled manner. If a model performs well on a corruption it has never seen, it can be said to be robust to this corruption. For example, ImageNet-C (Hendrycks & Dietterich, 2019), which introduces different parametric corruptions to the ImageNet validation set, has long stood as the de facto standard for OOD evaluation. Models that were trained on the (uncorrupted) ImageNet images must robustly generalize in order to perform well on ImageNet-C.

With the shift towards training models on vast, largely unfiltered image datasets, it is much less clear how to obtain test images that are truly OOD. Adding noise and blur to images can no longer be considered a distribution shift, because such images are already present in the training set, as demonstrated in Fig. 1. Models trained on LAION have seen the types of corruption in ImageNet-C, and are likely exposed to all realistic corruptions. Therefore, the fact that models like CLIP (Radford et al., 2021) exhibit much better performance on classic OOD datasets than ImageNet-trained models might not be an indication of true OOD robustness, but rather a consequence of a smaller train-test gap. For distribution shifts defined by the style of an image, recent work empirically shows that such datasets are indeed not OOD, but overlap with LAION-400M (Mayilvahanan et al., 2023; 2024). Hence, to measure the OOD robustness of modern models, a dataset containing truly new image corruptions is needed—even if this means that the corrup-

---

[‡]Supervised this work. [1]Max Planck Institute for Intelligent Systems, Tübingen, Germany [2]ELLIS Institute Tübingen [3]Tübingen AI Center [4]Google DeepMind. Correspondence to: Fanfei Li <first.last@tuebingen.mpg.de>.

tions must be highly artificial.

Previous work (e.g., Hendrycks & Dietterich, 2019) found that OOD generalization is not trivial to achieve: Many vision models do indeed struggle with OOD datasets like ImageNet-C, even if they perform well on ImageNet itself. Hence, these types of unfamiliar inputs are crucial for evaluating the robustness of machine learning models since they are indicative of performance on unexpected input; a challenge that many deployed models face.

Given the importance of OOD generalization in practice, there is a pressing need for a new benchmark that more effectively evaluates the OOD robustness of state-of-the-art foundation models: an OOD dataset for the era of web-scale vision models. Our **contributions** are as follows:

1. Given that existing OOD datasets are often no longer OOD for models trained on web-scale datasets, we introduce LAION-C, a **novel benchmark** dataset with six manually designed corruption types and 16 superclasses to evaluate the robustness of web-scale vision models. This dataset serves as a proxy for unseen challenges, allowing us to probe the limits of current models' robustness in a controlled but challenging environment.

2. We conduct a comprehensive performance analysis of various models on LAION-C and report a robust human OOD generalization baseline obtained through **psychophysical experiments** with 19 participants, collecting 11,400 trials in a highly controlled laboratory environment.

3. The resulting data serves as an OOD benchmark for current and future models, enabling not only an assessment of their generalization ability on truly OOD data but also providing insights into the **discrepancies between human and machine perception**, observing a paradigm shift in OOD generalization: from humans outperforming models to the best models now matching or outperforming the best human observers.

A detailed related work section can be found in Appx. A.1.

## 2. Methods

### 2.1. Constructing New OOD Distortions

As described in the introduction and depicted in Fig. 1, ImageNet-C does no longer qualify as out-of-distribution (OOD) for models trained on large-scale datasets and, therefore, can no longer be employed for such testing. Given the limitations of existing benchmarks like ImageNet-C, we develop a novel dataset specifically designed to challenge these foundation models more rigorously. We do this by introducing a covariate shift, where

$P(X)$ changes, while $P(Y|X)$ remains the same (as opposed to concept drift or label shifts, commonly referred to as "semantic" shifts (Yang et al., 2024)). Our dataset introduces six carefully designed, fully synthetic distortions that models have not encountered during training. These distortions are designed to be OOD even for web-scale datasets (as supported by quantitative evidence presented later). Hence, models truly need to generalize beyond their training distributions to perform well on this benchmark which we term LAION-C.

**Distortions** The core idea behind our distortions is to create highly synthetic, artificial corruptions that do not naturally appear even in large-scale datasets and are therefore OOD even for modern vision models. To achieve this goal, we intentionally disrupt visual consistency and perceptual cues that models might rely on for image classification. We began with a diverse pool of potential distortions, each designed to target different aspects of visual processing: texture handling (Geirhos et al., 2019), color perception, or edge detection. Initial testing revealed that while contemporary models are adept at navigating simpler distortions, their performance varies significantly with more complex challenges. This insight prompted us to carefully choose distortions that provide a sufficient challenge to the models while also offering varied levels of perceptual difficulty for humans.

Our final distortions are selected to fulfill two key desiderata: (1) they are exotic enough to have a low occurrence probability even in web-scale datasets, and (2) they test feature extraction capabilities relevant to robust object recognition. For instance, solving the *Stickers* and *Mosaic* distortions requires a model to holistically integrate the image, rather than being misled by the local image cues introduced by sub-images—a scenario deep neural networks notoriously struggle with (Geirhos et al., 2019). The *Glitch* and *Vertical Lines* distortions are among the most exotic and globally disruptive transformations we could construct, as they effectively destroy the texture cues that many models rely on (Geirhos et al., 2019). The *Geometric Shapes* distortion tests amodal completion, a fundamental aspect of human visual processing observable even in infants (Kellman & Spelke, 1983; Nanay, 2018), and also alters the image's color distribution—something humans are robust to, as we do not primarily depend on color for object recognition (Biederman & Ju, 1988; Tanaka & Presnell, 1999). Lastly, the *Luminance Checkerboard* distortion tests a model's ability to adapt to local lighting conditions, an essential ability of the human visual system (Carandini & Heeger, 2012). This way, LAION-C complements existing benchmarks by including images that challenge human perception, instead of limiting ourselves to visual domains in which humans excel.

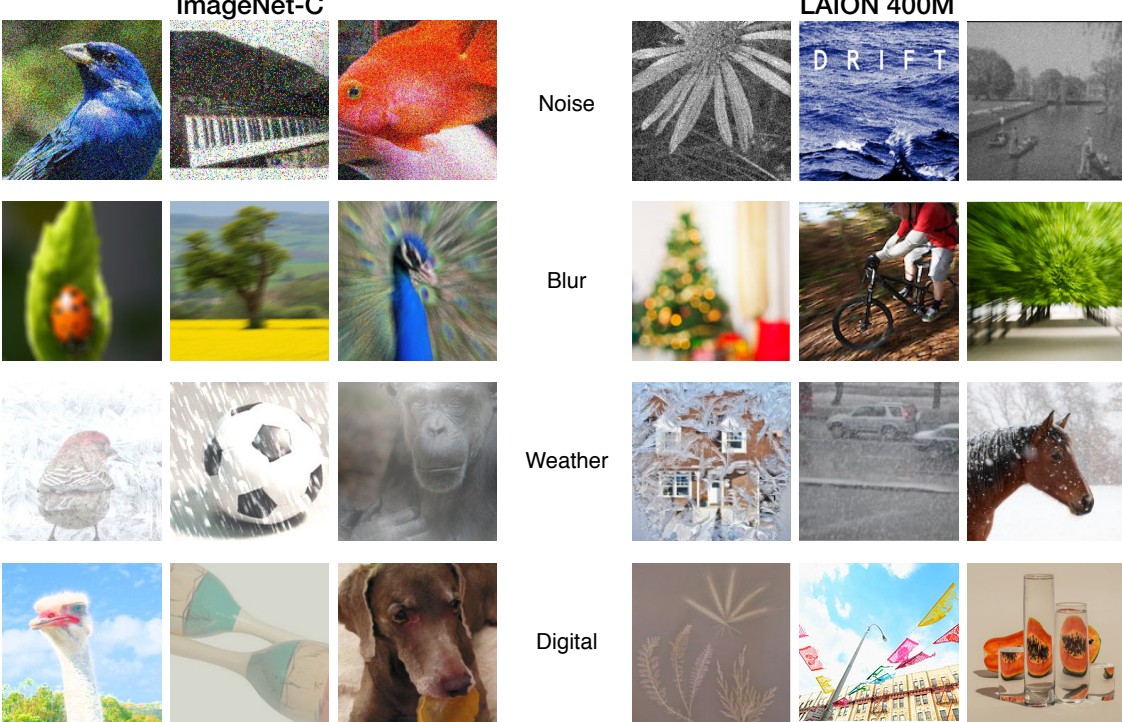

*Figure 1.* **ImageNet-C corruptions are not out-of-distribution (OOD) for web-scale datasets like LAION-400M.** Exemplary corrupted images from ImageNet-C (left) are similar to LAION-400M samples (right). Each row shows example corruptions and dataset images for one ImageNet-C corruption category (Noise, Blur, Weather, Digital). The presence of these distortions in web-scale datasets indicates the need for an OOD benchmark in the era of web-scale vision models.

As AI systems are increasingly deployed in complex and high-stakes domains, it is crucial that benchmarks evolve to robustly test these systems' generalization capabilities. Following ImageNet-C, each distortion consists of five different *intensity levels*. The distortions capture a range of visual challenges, as described below and illustrated in Fig. 2.

- **Mosaic:** The original image is broken down into smaller tiles, each replaced by a chromatically similar picture. This patchwork creates a mosaic effect that disrupts edges and textures while introducing contextually irrelevant information.

- **Glitched:** The original image undergoes an artistic digital corruption with horizontal lines overlaying shifted image segments and color channel shifts. This dislocates the global contextual structure of the image. While the concept of such glitchy images has been explored in earlier work (Kaufmann et al., 2019), our transformation introduces a more intense corruption.

- **Vertical Lines:** The original image is deconstructed into bent vertical line segments. This distortion retains the original colors but strips away local information, disrupting the contours and edges of the image and introducing visual discontinuity.

- **Geometric Shapes:** The original image is overlaid with overlapping geometric figures such as squares, circles, and stars. This visual clutter introduces local noise that obscures the main object, like the Kaleidoscope corruption from (Kaufmann et al., 2019).

- **Stickers:** The original image is augmented with assorted image patches. This addition of visual elements masks features of the primary object.

- **Luminance Checkerboard:** The original image is divided into a grid, with the luminance of each cell altered in a checkerboard pattern. The stark luminance contrast between adjacent tiles and artificial grid boundaries makes this distortion challenging.

We intend to build a challenging dataset that has the potential to guide the future development of vision models. Our dataset incorporates these tougher and less common distortions to simulate the difficulty of OOD scenarios that models might encounter in real-world applications. In line with established psychophysical experimental paradigms, we calibrate the intensity levels of our distortions to range from near-perfect to chance-level difficulties. We tune these levels such that either humans or a contemporary vision model (ViT-B) achieve chance performance on the

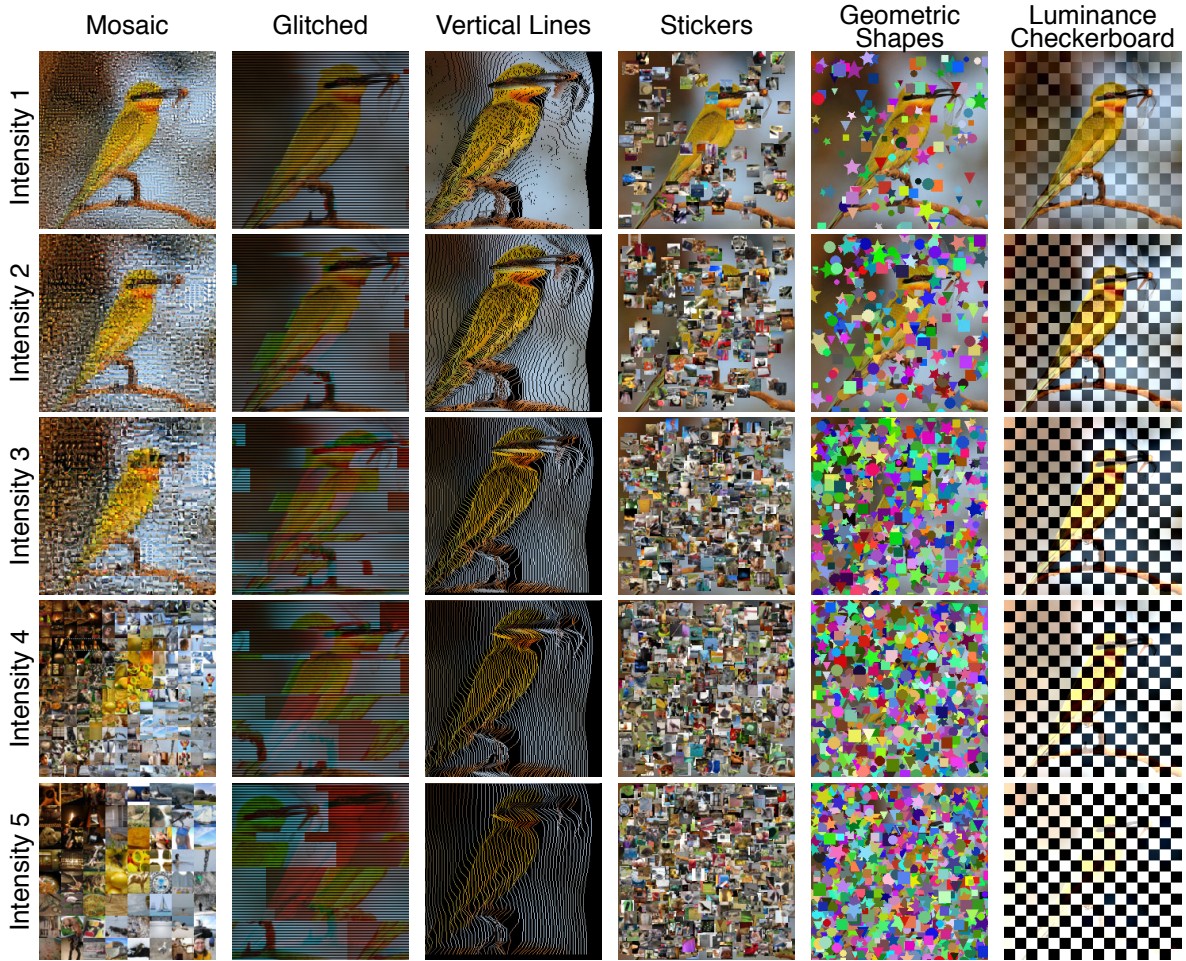

*Figure 2.* **LAION-C distortions, intended to be OOD even for web-scale datasets.** This figure illustrates the six LAION-C distortions at five intensity levels. Following the standard experimental paradigm from psychophysics, our dataset spans from near-perfect to chance-level difficulties, thoroughly testing models and leaving room for future model improvements. Best viewed on screen.

highest intensity level, i.e. no model is expected to perform well on the hardest levels. The other intensity levels are chosen so that we can observe a gradual decline in accuracy, ensuring that the distortions are sufficiently challenging.

These distortions are then applied to a carefully curated subset of images from the ImageNet validation dataset. To contextualize model performance, we intend to compare it to human performance. As human evaluations on datasets with hundreds of classes cannot be scaled to sufficiently many participants, we follow previous work (Geirhos et al., 2018) and simplify the task to a 16-class classification problem. We extract 285 ImageNet-classes to form 16 superclasses, namely ball, bird, boat, bottle, butterfly, car & truck, cat, chair, dog, fish, fruit, instrument, primate, snake, timekeeping, and tool. For robust statistical analysis, our dataset comprises 273 images for each superclass, each of which is corrupted at 5 intensity levels for all 6 distortion

types, resulting in $> 130k$ total images. This data size selection allows us to ensure that a 3% difference in the performance between models is statistically significant. Additionally, we manually filter the dataset to ensure that none of the images in one superclass contain objects from another class or require specific cultural knowledge for classification. This process helps to avoid ambiguous ground-truth labels.

## 2.2. Measuring Model Performance

We use the generated datasets to evaluate the performance of a suite of 58 vision models. Our selection includes models trained on large-scale web datasets and fine-tuned on ImageNet-1k, such as Vision Transformers (ViT) (Dosovitskiy et al., 2021), ConvNeXt (Liu et al., 2022), and EVA (Fang et al., 2023; 2024). For comparison, we also evaluate the performance of smaller-scale model families such as ResNet (He et al., 2016) and MobileNet (Howard,

2017) and large-scale models trained only on ImageNet-1k. Additionally, we also evaluate GPT-4o (OpenAI, 2024) and Gemini 1.5 Pro (Team et al., 2024) on a representative subset of LAION images. See Tab. 11 for a complete list of all models we evaluate. To address the imbalance caused by distinct numbers of subclasses within each superclass, we compute the average probability values across subclasses for each superclass, a method first suggested by (Geirhos et al., 2018). This adjustment mitigates biases introduced by the varying subclass distributions, ensuring a more accurate model performance evaluation.

### 2.3. Collecting Human Performance via Lab Experiments

To explore the discrepancies between human and machine perception, we design a psychophysical experiment to gather human classification data on the augmented images. This experiment builds on previous paradigms (Geirhos et al., 2018; 2021) to ensure consistency and comparability. In the experiment, 19 human subjects are briefly presented with a distorted image and are asked to classify it into one of 16 classes, reminiscent of how a DNN might be evaluated on a classification task (in contrast to e.g. open response paradigms, where participants could give arbitrary textual responses). Participants were recruited from the university student body, and screened for normal or corrected-to-normal vision and absence of color blindness. The experiment was conducted in a controlled dark environment using a 22" VIEWPixx 3D light LCD monitor, with stimuli presented centrally at a fixed viewing distance to ensure foveal presentation. The classification task was divided into two warm-up blocks and ten main experiment blocks, with each block containing images from 16 superclasses. Participants were given 2.5 s to view each image, followed by a 2 s response window to classify the image by clicking on a set of icons. To motivate high performance, a monetary bonus was awarded for surpassing fixed, predetermined performance thresholds for each block. Further methodological details are provided in Appx. A.2.

### 2.4. Quantifying Human-Machine Alignment via Error Consistency

To quantify the alignment between human and machine visual perception, we adopt the error consistency metric proposed in Geirhos et al. (2020b). Error consistency, denoted as $\kappa \in [-1, 1]$, measures the degree of agreement between the classification mistakes of two observers. In brief, $\kappa$ takes on a value of 1.0 if two observers are perfectly consistent, i.e. if they make classification mistakes on exactly the same images. Two independent binomial observers that agree no more than expected by chance will result in a $\kappa$ of 0, while two maximally inconsistent observers will have a $\kappa$ of -1. See Appx. A.3 or Geirhos et al. (2020b) for a detailed explanation of the metric.

## 3. Results

### 3.1. How OOD is LAION-C?

Next, we empirically evaluate whether our LAION-C dataset is indeed OOD relative to the large-scale image datasets used to train modern vision models. Rigorously quantifying how OOD a test dataset is with respect to some training dataset requires a precise definition of the test and training domain (Mayilvahanan et al., 2024). As the distribution shifts introduced by the distortions of our proposed LAION-C and ImageNet-C are fuzzy in nature, we use three tools to compare the OOD-ness of our proposed dataset to the OOD-ness of ImageNet-C. First, we use a qualitative assessment. By searching for the name and related concepts of ImageNet-C corruptions, we easily find LAION samples with visual distortions akin to those of ImageNet-C samples (see Fig. 1).

Second, we use the difficulty of a test dataset (measured by the performance that models trained on a reference dataset yield on the test dataset) as a proxy for how much the test dataset differs from the reference dataset. Here, the reasoning is that if a test dataset can be solved almost perfectly by a model, it means that either the model has great generalization skills or the test dataset is not strictly OOD. If, at the same time, another dataset has much greater difficulty according to the same models, the second dataset is likely more OOD than the first. For the sake of comparability, we here use a version of ImageNet-C restricted to the same 16 superclasses that were used for LAION-C, where we implemented the ImageNet-C augmentations through the code by Michaelis et al. (2019). Indeed, a comparison of the performance achieved by our suite of models (see Fig. 3) suggests that LAION-C is more OOD to LAION than ImageNet-C is.

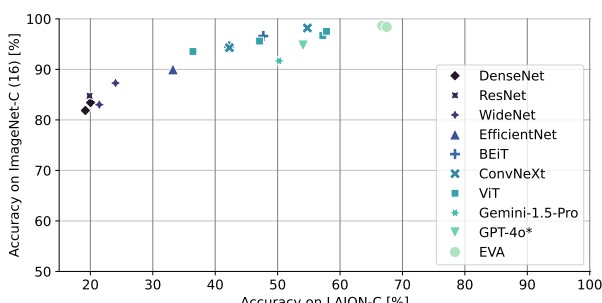

*Figure 3.* **Performance Divergence of Models on LAION-C and ImageNet-C 16 class.** Evaluating models on the 16-class versions of ImageNet-C and LAION-C produces a plateaued performance on ImageNet-C, while LAION-C still yields a high variance across models.

Third, we use the FID (Heusel et al., 2017; Kynkäänniemi et al., 2022) to quantify the difference between LAION and ImageNet-C and LAION-C, respectively. Specifically, we employ a CLIP-trained ViT-B as feature encoder and use the implementation by Parmar et al. (2022) to calculate FID-scores. In line with the previous evidence, we find a FID of $\approx 70$ between LAION and LAION-C, which is substantially higher than that between LAION and ImageNet-C ($\approx 40$). This means that features of LAION are closer to those of ImageNet-C than those of LAION-C, again highlighting the greater OOD-ness of LAION-C. In summary, we have presented three different kinds of evidence suggesting that LAION-C is more OOD than ImageNet-C to LAION.

### 3.2. Comparison to other OOD benchmarks

We conduct a direct comparison between LAION-C and other well-established out-of-distribution (OOD) datasets. As illustrated in Figure 4, LAION-C provides a more detailed resolution of model performance variances. This dataset captures a broader variance in model performance, with a standard deviation of approximately ~27%, compared to an average of ~10% in other common OOD datasets. Notably, LAION-C is evaluated on a 16-class basis, which is significantly fewer than the 200-1000 classes used in typical OOD datasets, making the results even more remarkable.

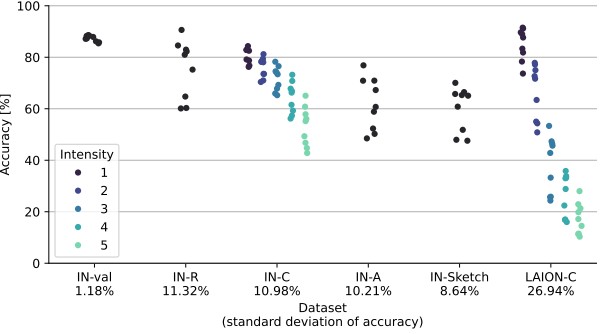

*Figure 4.* **LAION-C offers better resolution of model differences.** We tested 9 models pre-trained on LAION2B, evaluating them across all intensity levels if applicable. LAION-C captures a broader variance in model performance, with a standard deviation of ~27%, compared to an average of ~10% in other common OOD datasets. Notably, LAION-C is tested on a 16-class basis, while other datasets typically use 200-1000 classes, making this result even more remarkable.

### 3.3. Machine Performance

In Fig. 5, we compare model performance on ImageNet-C against performance on LAION-C. Evidently, the average model performance on ImageNet-C stays above or close to

60%, indicating that current models are increasingly adept at handling the distortions in ImageNet-C. This observation reinforces our hypothesis that the challenge presented by ImageNet-C may no longer be sufficiently difficult to rigorously test the robustness of modern models.

In contrast, models achieve much lower accuracy on LAION-C on average and exhibit more inter-model variability. This showcases our dataset's ability to uncover nuances that remain hidden on more saturated benchmarks. These performance differences are particularly obvious at higher intensity levels, illustrating LAION-C's potential to serve as a more challenging and insightful benchmark for evaluating robustness.

We also provide a detailed breakdown of non-averaged, dataset-specific results in Fig. 11. We observe significant variability in the performance of different vision models across various datasets and distortion levels, highlighting the diversity in model robustness. These results further highlight the effectiveness of our datasets in eliciting different responses from models of different architectures, parameter scales, and training data sizes. This diversity is particularly valuable for understanding which model designs are more robust to specific types of distortions, offering insights that are beneficial for advancing the state-of-the-art model robustness.

In Tab. 10 in appendix, we present a comprehensive evaluation of our suite of models on LAION-C. We report each model's top-1 accuracy on the (undistorted) ImageNet validation set as a baseline (*Clean Accuracy*) and the average top-1 accuracy on LAION-C averaged across distortion types and intensity levels (*LAION-C*). We then break the latter down into the six distortion types. This enables a thorough comparison of model performance, highlighting which architectures generalize best.

### 3.4. Can LAION-C Be Solved?

Given the low performance of current state-of-the-art models on LAION-C, one might wonder whether LAION-C is simply impossible to solve because the distortions destroy all information necessary for correct classification of the images. To disprove this hypothesis and highlight the validity of LAION-C as a benchmark for evaluating model robustness, we conduct a fine-tuning experiment to assess whether the challenges posed by LAION-C are solvable at all. Specifically, we fine-tune a ViT-Huge model, which was originally pretrained with a CLIP-objective on LAION-2B and then fine-tuned on ImageNet-22k and ImageNet-1k. For this experiment, we use a custom dataset sub-sampled from the ImageNet-1K training set and augmented with the distortions introduced in LAION-C. This dataset consists of over 336,000 images uniformly sampled across the 16 superclasses defined for LAION-C.

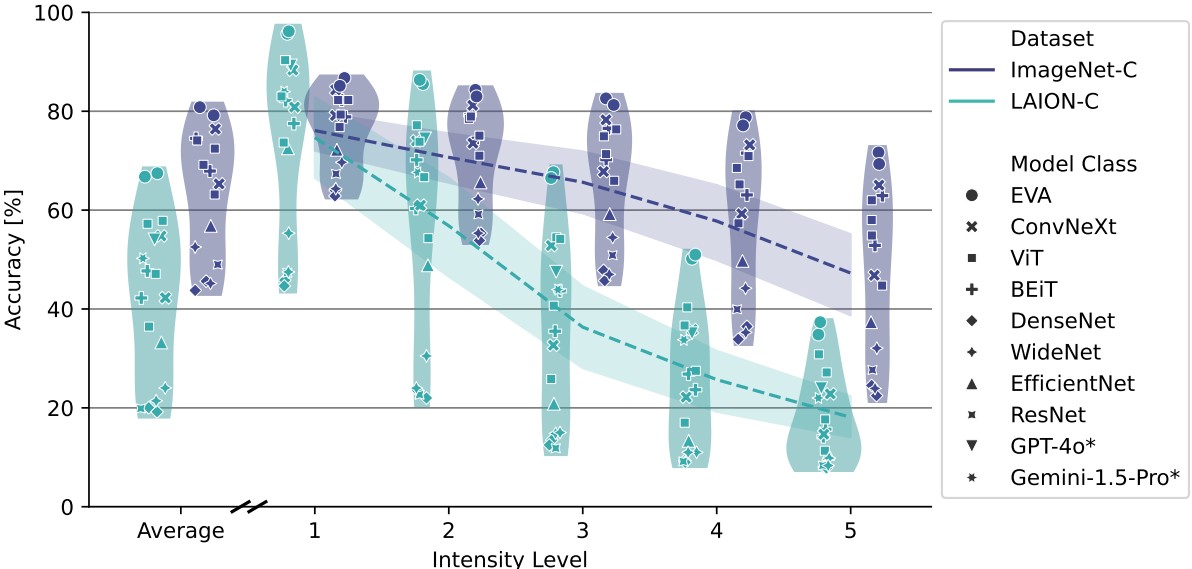

*Figure 5.* **LAION-C poses a greater challenge to model robustness than ImageNet-C.** We plot distortion intensity against each model's average accuracy. Visual foundation models evaluated on ImageNet-C maintain high accuracy, with minimal drop across increasing intensity levels. On our LAION-C dataset, the models exhibit a sharper decline in accuracy, highlighting the benchmark's effectiveness in measuring model robustness.

*Table 1.* **LAION-C is challenging but can be solved by fine-tuning on the exact distortions.** We compare the performance of ViT-H-P14-336-CLIP-LAION-IN12K before and after fine-tuning it on ImageNet-1k training images with LAION-C corruptions. As the performance after fine-tuning is high, this means that LAION-C, although challenging, remains solvable as it retains enough signal when applying distortions.

| Accuracy [%] | Mosaic | Vertical Lines | Glitched | Luminance | Geometric | Stickers | ImageNet-Val (16 class) |
|---|---|---|---|---|---|---|---|
| Before | 45.2 | 51.2 | 69.8 | 88.2 | 64.4 | 24.6 | 99.8 |
| After | 80.6 | 93.6 | 96.8 | 97.8 | 89.8 | 67.4 | 99.2 |

As shown in Tab. 1, fine-tuning the model results in substantial accuracy gains, which define an upper bound on LAION-C accuracy that no normal model can be expected to achieve. Notably, these accuracy gains are particularly pronounced on higher-intensity distortions, as detailed in Tab. 9, where accuracy is broken down by distortion intensity. The fine-tuned model likely achieves such good performance by employing un-human-like (or "spurious") features, but the purpose of this experiment is *not* to suggest that fine-tuning on LAION is a sensible approach to improve OOD robustness, but to quantify how much learnable signal is left. LAION-C provides meaningful robustness tests without being intractable, making it a valuable tool for a more comprehensive evaluation of model performance under difficult conditions.

## 3.5. Human and Machine Vision Discrepancy

**Accuracy Differences.** In Fig. 6, we summarize how our suite of models performs in terms of classification accuracy, compared to the human participants in our psychophysical experiment. We report the best performances, since averages would be unfairly influenced by some older models we included as points of comparison. In Fig. 11, we provide a more detailed breakdown of performance by model. While human observers still outperform most vision models on images with Mosaic or Glitched distortions, the best models match (or even slightly surpass) human performance. For distortions involving occlusion and luminance manipulations, the vision models typically achieve higher accuracy than humans. Overall, current state-of-the-art vision models now match or even outperform human observers in OOD scenarios, including on our synthetic distortions, which they likely have never encountered during training—a stark contrast to just a few years ago, when humans were still vastly outperforming models (Geirhos et al., 2018; Dodge & Karam, 2019; Taori et al., 2020; Jang & Tong, 2024).

**Performance on Complex Distortions.** When analyzing more complex distortions such as Mosaic, Vertical Lines, and Glitched images (first row of Fig. 6), we observe that human performance is generally on par with the best-performing models. Especially at greater intensity levels, humans perform competitively, e.g., outperforming

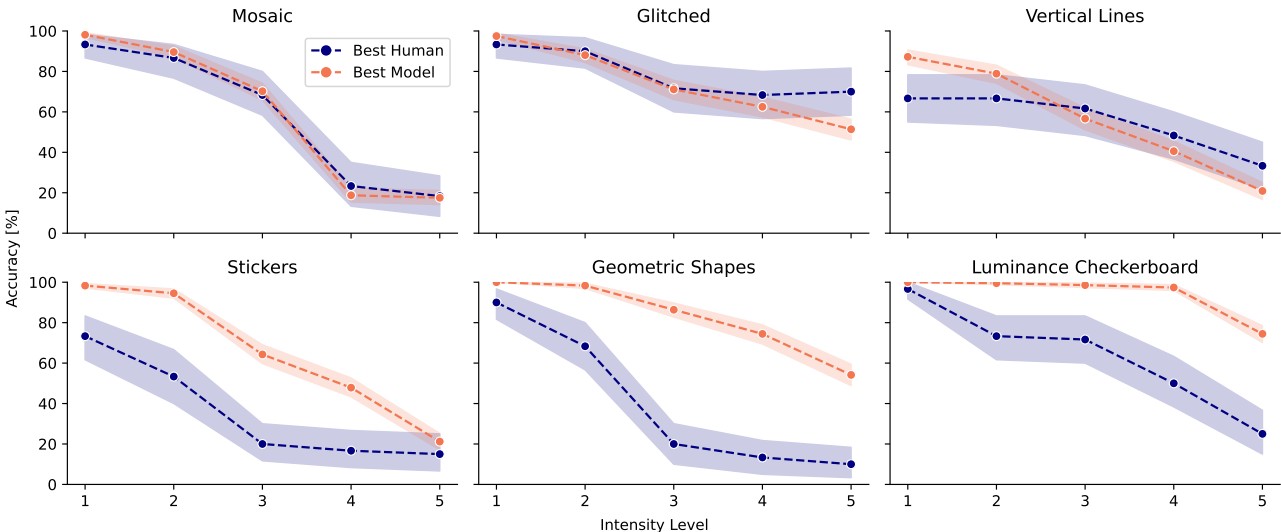

*Figure 6.* **Human vs. machine accuracy on all distortions.** For each LAION-C distortion, we plot the distortion intensity against the accuracy of the best human and the best model in this condition (for average human performance, see Fig. 11). The shaded regions indicate the 95% confidence intervals around the means. On the Mosaic, Glitched and Vertical Lines distortions, humans and machines perform similarly, whereas the best model vastly outperforms the best human observer on the Stickers, Geometric Shapes, and Luminance Checkerboard distortions.

all models for the strongest Vertical Lines distortions. As we show in Fig. 11, the gap between humans and older models like the ResNet variants is particularly large on these complex distortions. However, modern model classes demonstrate substantial progress, approaching human-level performance even at higher intensity levels. While some margin for improvement remains, the narrowing gap suggests that achieving human-level robustness on classification tasks is no longer the primary challenge for state-of-the-art architectures.

**Human-Machine Error Consistency on LAION-C.** For a more fine-grained analysis of the behavioral agreement between models and human observers, we calculate error consistency as described in Sec. 2. As illustrated in Fig. 8 in appendix, there is a high degree of variability in error consistency between human observers and different vision models ranging from 0 to 0.4. This indicates that while model families such as ViT and EVA rival or surpass human performance, they are approaching the task utilizing different strategies than humans, demonstrating less human-like behaviors. The observed value range matches the one found in previous work for older models and different image data (Geirhos et al., 2021). This again suggests that while recent developments have boosted model performance, these models have not become more human-like, as they follow alternative strategies.

## 4. Discussion

**Summary.** Given that existing OOD benchmarks are often no longer OOD for models trained on web-scale datasets like LAION since distortions like blur and digital corruptions are commonplace on the web, we here introduce LAION-C. LAION-C is a benchmark designed to evaluate the robustness and generalization capabilities of modern vision models trained on web-scale datasets. Our empirical results demonstrate that LAION-C is more challenging for a representative suite of vision models than the existing ImageNet-C benchmark, particularly at higher distortion intensity levels. Our thorough human evaluation in a highly controlled psychophysical laboratory totaling 11,400 trials shows that *the best models often outperform even the best human observers*. While they do not always follow similar strategies (as indicated by error consistency analysis), this reassuring finding indicates that the best models have indeed substantially progressed in their ability to handle unexpected input and are not just getting better on in-distribution distortions. Given that the LAION-C benchmark dataset, by virtue of its construction, serves as a better proxy for a model's ability to recognize objects despite an unexpected distortion, we recommend it as an OOD benchmark for current and future web-scale vision models.

**Limitations.** While we have shown that LAION-C can effectively reveal shortcomings in model robustness, we have not yet fully explored why certain models underper-

form on specific distortions. Although our empirical results provide valuable insights, further investigation is required to clarify which visual cues the models rely on under different conditions. Such an analysis could inform the development of new inductive biases or architectural improvements, since a better understanding of these mechanisms could lead to improvements in both model interpretability and robustness. Given our current focus on introducing the dataset, this was not fully addressed here, but could be an area for future exploration. Furthermore, it is an open question what the performance limit on LAION-C looks like. Since fine-tuning models on LAION-C results in significant performance gains, particularly at higher distortion levels, there clearly is potential for optimization through advanced training techniques. However, how to further improve generalization across OOD scenarios, especially to enhance the models' ability to handle the novel distortions presented by LAION-C, remains an open question that warrants further exploration. To retain its value as an OOD benchmark, LAION-C should not be used as a training or fine-tuning dataset (except for analysis purposes).

**Conclusion and outlook.** Just a few years ago, early investigations into generalization abilities of deep neural networks showed humans vastly outperforming the best models (Geirhos et al., 2018; Dodge & Karam, 2019). Fast-forwarding to today, LAION-C shows that the best models either match or outperform human performance on challenging OOD distortions. This finding is reassuring in the light of growing concerns about the quality of existing evaluation datasets, including the concern that OOD datasets like ImageNet-C may no longer serve their original purpose in the era of web-scale training datasets. Our findings indicate that the often *super-human performance* of modern models is achieved through *super-human strategies*: Models use a variety of image cues—including, in all likelihood, background pixels to perform well on some of our challenging datasets. Given their high performance across the board, they no longer rely on a single strategy that fails when faced with a test case that distorts a particular image cue. This marks a paradigm shift in OOD generalization: From humans outperforming models to models outperforming humans, from relying on a single strategy to a diverse set of multiple robust strategies, and from OOD benchmarking measuring progress towards human-like object recognition to better performance now indicating super-human (in other words, *less human-like*) vision models.

## Impact Statement

We confirm that all experimental procedures involving human subjects in our study had IRB approval. All participants gave informed consent prior to their inclusion in the study. Detailed information was provided to each participant beforehand, outlining the study's purpose, procedures and benefits, ensuring they were fully informed before agreeing to participate. Participants were also informed that they could abort the study at any time, without incurring any negative consequences. Experimental data and contact information for the participants was stored in accordance with GDPR. Participants were compensated with an hourly base rate of 12 EUR and received bonus payments based on classification performance, as is customary in psychophysical experiments, so that the final reimbursements exceeded the local minimum wage. This paper presents work whose goal is to advance the field of Machine Learning. While we do not foresee any immediate negative societal consequences, the use of human participants highlights the importance of maintaining high ethical standards. To enhance transparency and reproducibility, we also provided a detailed datasheet outlining the dataset's characteristics, collection methods, and intended use cases.

## Code and dataset availability

The evaluation code for LAION-C is publicly available at: https://github.com/FanfeiLi/LAION-C. The LAION-C dataset is published on Zenodo. A link to the dataset is provided via the GitHub repository.

## Acknowledgements

We would like to thank Felix Wichmann for providing access to his psychophysics laboratory and valuable guidance, as well as the students who took part in our study. We also thank multiple anonymous reviewers for their helpful feedback on the manuscript. This work was supported by the German Federal Ministry of Education and Research (BMBF): Tübingen AI Center, FKZ: 01IS18039A. WB acknowledges financial support via an Emmy Noether Grant funded by the German Research Foundation (DFG) under grant no. BR 6382/1-1, via the Open Philanthropy Foundation funded by the Good Ventures Foundation and the German Research Foundation (DFG) through the SFB 1233, Robust Vision: Inference Principles and Neural Mechanisms, TP A2, project number: 276693517. WB is a member of the Machine Learning Cluster of Excellence, EXC number 2064/1 – Project number 390727645. This research utilized compute resources at the Tübingen Machine Learning Cloud, DFG FKZ INST 37/1057-1 FUGG. The authors thank the International Max Planck Research School for Intelligent Systems (IMPRS-IS) for supporting Thomas Klein.

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

# A. Appendix

## A.1. Related Work

**OOD generalization ability of vision models.** As deep learning has advanced to the point where models can reliably generalize to data that matches their training distribution or even exceed the quality of the original labels (Beyer et al., 2020), OOD-robustness, as measured by specifically designed OOD test sets, has moved to the center stage of computer vision research. In particular, ImageNet-C (Hendrycks & Dietterich, 2019), a dataset containing images from the test-set of ImageNet to which various fairly natural corruptions such as blurring and pixelation were applied, has long been the gold standard for assessing OOD-performance, to the point where data augmentations proposed to increase OOD robustness were found to only work well because they are similar to the ImageNet-C corruptions (Mintun et al., 2021). In contrast, ImageNet-R (Hendrycks et al., 2021a) instead provides a more complex distribution shift by collecting different renditions of the target classes such as sculptures and paintings, instead of photos. In a similar vein, (Wang et al., 2019) introduce a dataset of black and white sketches matching the labels and scale of the ImageNet validation set, called ImageNet-Sketch. A more subtle distribution shift which still caused considerable drops in model performance for ImageNet-trained models, was proposed by Recht et al. (2019). They collected ImageNetV2, a new test set for ImageNet that should theoretically not differ from the ImageNet test set at all, because it was collected with a very similar methodology, but revealed that models do perform slightly worse on ImageNetV2 than on the original test set. Hendrycks et al. (2021b) proposed two other OOD-test sets which do not rely on synthetic image manipulations but instead consist of natural images that are in some sense OOD relative to ImageNet, either by virtue of displaying object classes not present in ImageNet (ImageNet-O) or by showing an object of an ImageNet-class in a scene that is weird enough to fool most models (ImageNet-A). What all of these datasets have in common is that, by design, they provide distribution shifts *relative to ImageNet*. But with the rapid evolution of deep learning, these traditional benchmarks have become less challenging for state-of-the-art vision models trained on expansive web-scale datasets (Radford et al., 2021). While it is to some degree possible to predict a model's OOD generalization from its training set performance (Taori et al., 2020), the only reliable measurements of this capability stem from empirical evaluations of models on OOD datasets. Our work addresses this need by introducing LAION-C, a dataset that incorporates novel and complex synthetic distortions tailored to challenge even advanced vision systems.

**Advancement in visual foundation models** The rise of visual foundation models such as Vision Transformers (ViT) (Dosovitskiy et al., 2021), ConvNeXt (Liu et al., 2022) and BeiT (Bao et al., 2022) has redefined what constitutes standard performance across many visual tasks. These improvements in performance partially stem from architectural innovations and parameter optimization, but were mostly powered by the effective leveraging of unprecedented dataset sizes (Zhai et al., 2022). However, because visual foundation models were trained on web-scale datasets, the extent of their generalization capability remains underexplored.

**Comparing human vs. machine perception.** Deep Neural Networks were originally conceived as models of human vision (Fukushima, 1975) and were found to be the best available models for neuronal activity in the primate visual cortex (Yamins et al., 2014), even if not trained for this task. Today, there is a growing body of research dedicated to evaluating the adequacy of neural networks as behavioral models of human core object recognition (Doerig et al., 2023; Schrimpf et al., 2018; Wichmann & Geirhos, 2023; Muttenthaler et al., 2023). Building upon the findings of (Geirhos et al., 2021), who illustrate the narrowing of the behavioral gap between humans and machines in terms of their error consistency, our study further explores this dynamic utilizing LAION-C. We conducted a systematic analysis of differences in perception between human and machine observers, and assessed if the behavioral gap is closing further, as well as highlighting the persistent cognitive differences between humans and machines.

## A.2. Experiment Procedure and Participant Incentives

**Participant recruitment and setup.** We recruited 20 participants (10 female) from the university student body via mailing lists. All participants were screened to ensure normal or corrected vision and no color blindness, and gave informed consent to participate. One participant was excluded post-hoc due to reporting extreme tiredness. Our experiments were conducted in a darkened cabin, using a 22" VIEWPixx 3D light LCD monitor (VPixx Technologies, Saint-Bruno, Canada) at a refresh rate of 120 Hz (scanning backlight mode on). The screen measures $484 \times 302$ mm, at a resolution of $1920 \times 1200$ pixels. Stimuli were presented foveally in the center of the screen, with a viewing distance of 65 cm, resulting in 5 ° of visual angle. In line with earlier experiments, the background was set to a grey value of $0.454$ in the $[0, 1]$ range. A chin rest was used to maintain a fixed viewing distance and angle. The experiment was implemented using the Psychophysics

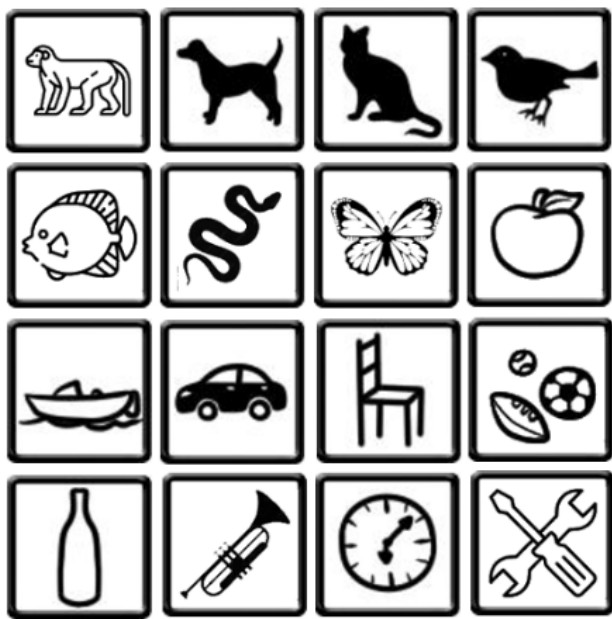

*Figure 7*. **Interface presented to participants.** This figure illustrates the icon layout as displayed to participants during the study. The grid is adapted from (Geirhos et al., 2018), while most of the categories and therefore symbols are different.

Toolbox (Kleiner et al., 2007, version 3.0.12) in MATLAB (Release 2016a, The MathWorks, Inc., Natick, Massachusetts, United States) using a 12-core desktop computer (AMD HD7970 graphics card "Tahiti" by AMD, Sunnyvale, California, United States) running Kubuntu 14.04 LTS.

The entire classification task, including both the warm-up and main experiment phases, was organized into 12 blocks. In each block, participants were shown images from the 16 superclasses for 2.5 seconds—a duration empirically determined to balance efficient overall experiment length with sufficient exposure time allowing for viewing and consideration time. After each image, the 16 corresponding class icons were displayed on screen, allowing participants 2 seconds to classify each image into one of these categories. The icons were organized in a layout that roughly grouped them by size and general category (e.g., animals and objects), as illustrated in image Fig. 7. To encourage responses rather than leaving selections blank, a message was displayed at the top of the screen 0.75 second before icon display time ended, prompting participants to make a choice. At the end of each block, if a participant surpassed the 90% accuracy threshold calibrated using internal baseline performance data, they received an encouraging on-screen message ("Congratulations! You just earned some extra money!") along with a $0.50 bonus per block to incentivize higher performance.

**Warm-up session and main experiment.** The experiment began with a 10-minute warm-up session, allowing participants to familiarize themselves with the icon layouts and the classification task procedure through two practice blocks. Each practice block contained 45 images, with one block consisting of clean images and the other of augmented images. To avoid test-time adaptation, the augmentations used during the warm-up phase differed from those in the actual trials. The images used for the practice trials were also a subsample of the ImageNet validation dataset, but did not overlap with those used in the main experiment.

Following the warm-up, the main experiment proceeded consisting of 10 blocks, each block comprising 60 images. Each set of 5 blocks was augmented using a consistent method, with a balanced distribution across different intensity levels and superclasses. To avoid fatigue, participants were allowed an unlimited amount of time to rest between blocks and encouraged to rest their eyes or accommodate elsewhere.

## A.3. Error Consistency

Here, we provide a more detailed explanation of error consistency (EC), summarizing (Geirhos et al., 2020b). The EC between two classifiers which both classified a sequence of samples is obtained by first using the necessary ground-truth labels to assess which images each observer classified correctly. A trial increases the agreement if both classifiers solved it correctly, or if they both failed (and decreases it if only one of them got the trial correct while the other one failed). One then calculates how much more agreement was observed between the two classifiers than observers relative to the agreement expected by chance, relative to the maximum possible delta. This is done by calculating Cohen's Kappa (Cohen, 1960), which is defined as $\kappa = \frac{p_o - p_e}{1 - p_e}$, where $p_o$ is the (empirically measured) proportion of agreement-trials and $p_e$ is the (theoretical) expected agreement when modeling both observers as independent binomial observers. $\kappa$ takes on values between $-1$ and $1$, with a higher $\kappa$ signifying higher levels of agreement, and a $\kappa$ of 0 indicating that a pair of observers does not agree more frequently than one would expect by chance, given their marginal correctness probabilities.

In this work, we calculate the error consistency between model responses and human classification data. To do this, we first collect all human responses. Since each human participant saw a fresh set of stimuli, we thus obtain exactly one human response per image. We then calculate each model's EC to this list of human responses.

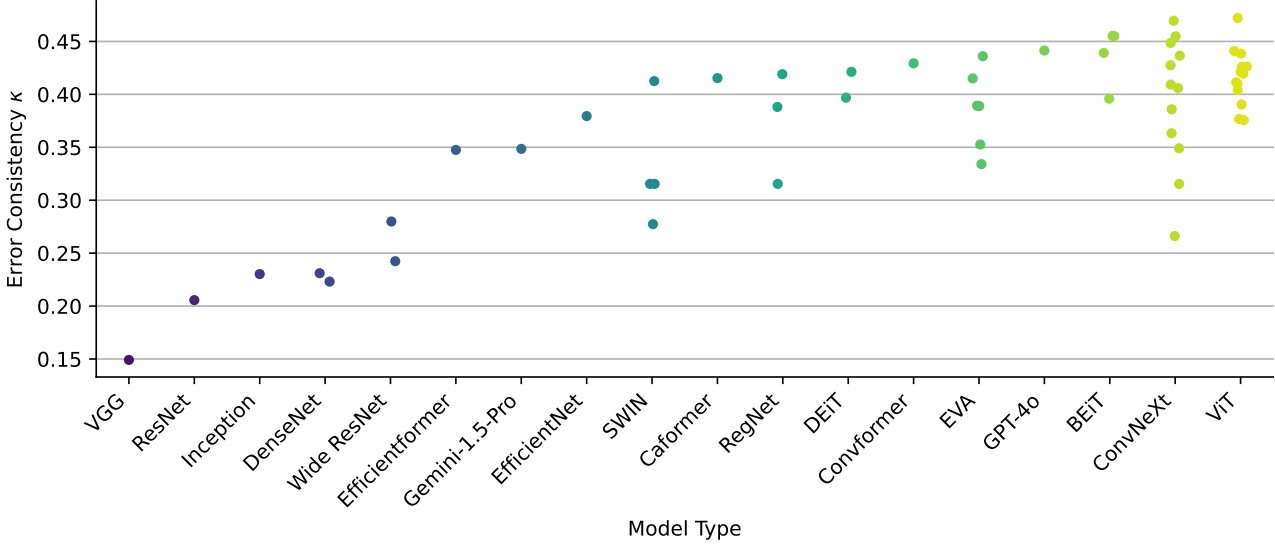

*Figure 8.* **Humans and models make different mistakes.** We analyze the agreement of error patterns between different families of vision models (see Tab. 11 for a complete list) and human observers. The error consistency ($\kappa$) could theoretically achieve a maximum value of 1, but in line with earlier work (Geirhos et al., 2021), the EC values range between 0.15 and 0.45, indicating that behavioral differences between humans and machines are still quite large. Marker colors encode model families.

## A.4. Augmentation Designs

### A.4.1. MOSAIC

The image is divided into an $n \times n$ grid, where each tile is replaced by a patch from the ImageNet validation set whose average color best matches that of the tile. This patchwork creates a mosaic effect that disrupts edges and textures while introducing contextually irrelevant information.

| Level | 1 | 2 | 3 | 4 | 5 |
|-------|---|---|---|----|----|
| **n** | 4 | 6 | 8 | 16 | 28 |

*Table 2.* Tile sizes at each level.

### A.4.2. GLITCHED

The original image undergoes an artistic digital corruption, with horizontal lines overlaying shifted image segments and color channel shifts. Here, a *region* refers to a randomly selected rectangular area of the image, and a *shift* denotes the horizontal displacement (left or right) of that region by a certain percentage of the image width. In addition, color channels are independently *offset* by a fixed number of pixels to further disrupt local spatial coherence. This dislocates the global contextual structure of the image. While the concept of such glitchy images has been explored in earlier work (Kaufmann et al., 2019), our transformation introduces a more intense corruption. Pixel shifts and color channel offsets are applied to random regions as follows:

| Level | 1 | 2 | 3 | 4 | 5 |
|-------|---|---|---|---|---|
| **Shift** | 8% width | 32% width | 50% width | 128% width | 200% width |
| **Regions** | 4 | 8 | 10 | 16 | 20 |
| **Offset** | $\pm 4$ px | $\pm 8$ px | $\pm 10$ px | $\pm 16$ px | $\pm 20$ px |

*Table 3.* Glitch parameters at each level.

The implementation is inspired by GitHub user "totallynotchase" (T, 2020).

### A.4.3. VERTICAL LINES

The original image is transformed through a process of vertical deconstruction. It is first divided into multiple vertical sections, which are further subdivided along the y-axis into small segments called y-steps. In each of these segments, a short vertical line is drawn, where the direction of each line subtly reflects the local contour or edge orientation of the image. The color of each line segment is set to the average color of its corresponding region. This distortion retains the original colors but strips away local information, disrupting the contours and edges of the image and introducing visual discontinuity.

| Level | 1 | 2 | 3 | 4 | 5 |
|-------|---|---|---|---|---|
| **Vertical Sections** | 224 | 178 | 112 | 84 | 60 |
| **Y-Step** | 1 px | 2 px | 4 px | 6 px | 8 px |

*Table 4.* Vertical sectioning and step sizes at each level.

### A.4.4. GEOMETRIC SHAPES

The original image is overlaid with overlapping geometric figures such as squares, circles, and stars. This visual clutter introduces local noise that obscures the main object, like the Kaleidoscope corruption from (Kaufmann et al., 2019). The number of shapes for each intensity level are shown as follows:

| Level | 1 | 2 | 3 | 4 | 5 |
|---|---|---|---|---|---|
| **Shapes** | 150 | 300 | 600 | 800 | 1000 |

*Table 5.* Number of shapes at each level.

### A.4.5. STICKERS

The original image is augmented by randomly placing $16 \times 16$ pixel image patches from the ImageNet validation set onto the image, following a uniform distribution. This addition of visual elements masks features of the primary object and introduces deceptive new features.

| Level | 1 | 2 | 3 | 4 | 5 |
|---|---|---|---|---|---|
| **Patches** | 100 | 200 | 400 | 600 | 1200 |

*Table 6.* Number of patches at each level.

For an estimate of the occlusion ration of the objects per intensity level for stickers and geometric shapes distortions, see Tab. 8.

### A.4.6. LUMINANCE CHECKERBOARD

The original image is divided into a $14 \times 14$ grid, with the luminance of each cell altered in a checkerboard pattern. The stark luminance contrast between adjacent tiles and artificial grid boundaries makes this distortion challenging.

| Level | 1 | 2 | 3 | 4 | 5 |
|---|---|---|---|---|---|
| **Luminance Variation** | $\pm 50$ | $\pm 50$–$100$ | $\pm 100$–$125$ | $\pm 125$–$150$ | $\pm 150$–$255$ |

*Table 7.* Luminance Variation at each level.

*Table 8.* **Occlusion ratio of objects in Stickers and Geometric Shapes distortions.** We calculated the object occlusion ratio for the Stickers and Geometric Shapes corruptions as an additional quantitative measurement of the distortion strength.

| Intensity Level | Geometric Shapes (%) | Stickers (%) |
|:---:|:---:|:---:|
| 1 | 61.88 | 65.83 |
| 2 | 72.51 | 76.52 |
| 3 | 85.35 | 86.19 |
| 4 | 90.16 | 89.54 |
| 5 | 93.21 | 91.63 |

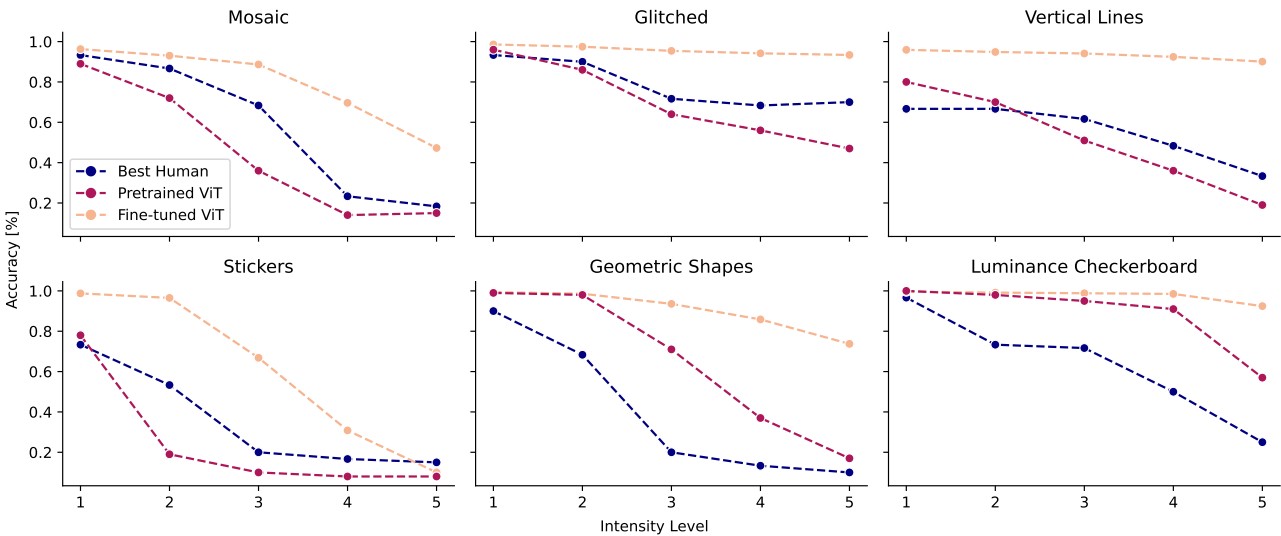

*Figure 9.* **LAION-C can be solved.** For every distortion, we plot the accuracy of our reference model (ViT-H-P14-336-CLIP-LAION-IN12K) before and after fine-tuning, in comparison to the best human participant for reference. Most distortions can be learned perfectly, only the Stickers and Mosaic distortions might have been too difficult at the highest intensity levels. Further performance gains might be possible with more careful fine-tuning.

*Table 9.* **Model (ViT) Accuracy Before and After Fine-Tuning on LAION-C.** The high accuracies after fine-tuning indicate that even though the dataset is challenging, there is, in principle, enough signal left to perform well on LAION-C.

|  | Intensity Level | Accuracy Before (%) | Accuracy After (%) |
| --- | --- | --- | --- |
| Mosaic | 1 | 89.0 | 96.3 |
|  | 2 | 71.9 | 93.0 |
|  | 3 | 35.8 | 88.7 |
|  | 4 | 14.3 | 69.6 |
|  | 5 | 14.7 | 47.7 |
| Vertical Lines | 1 | 79.9 | 95.9 |
|  | 2 | 70.1 | 94.9 |
|  | 3 | 50.8 | 94.1 |
|  | 4 | 36.1 | 92.4 |
|  | 5 | 19.4 | 90.0 |
| Glitched | 1 | 95.9 | 98.6 |
|  | 2 | 86.2 | 97.5 |
|  | 3 | 63.6 | 95.4 |
|  | 4 | 55.6 | 94.2 |
|  | 5 | 47.1 | 93.4 |
| Luminance Checkerboard | 1 | 99.7 | 99.6 |
|  | 2 | 98.4 | 99.2 |
|  | 3 | 95.1 | 98.8 |
|  | 4 | 90.7 | 98.5 |
|  | 5 | 56.6 | 92.5 |
| Geometric Shapes | 1 | 30.9 | 99.4 |
|  | 2 | 11.2 | 98.6 |
|  | 3 | 6.7 | 93.6 |
|  | 4 | 6.6 | 85.9 |
|  | 5 | 6.3 | 73.7 |
| Sticker | 1 | 97.3 | 98.8 |
|  | 2 | 77.8 | 96.5 |
|  | 3 | 28.7 | 63.7 |
|  | 4 | 14.9 | 31.8 |
|  | 5 | 8.1 | 14.3 |

## A.5. Breakdown of model performance

To demonstrate the value of LAION-C as a benchmark for evaluating model robustness, we analyze how model performance on LAION-C correlates with that on ImageNet-C. Grounding our comparison in models that have demonstrated a baseline level of robustness on well-established benchmarks, we apply a threshold to include 40 models that achieved at least 60% accuracy on ImageNet-C.

As shown in Fig. 10, the majority of data points lie above the identity line representing performance alignment on LAION-C and ImageNet-C. The gradual slope of the data points, combined with their positioning, indicates that models generally perform better on ImageNet-C, while their performance on LAION-C is more dispersed and often substantially lower.

This broader distribution of performance highlights that LAION-C introduces more challenging distortions, prompting models to exhibit greater variability in robustness. The moderate Kendall's tau coefficient ($\tau = 0.66$) between the models' performances on LAION-C and ImageNet-C further underscores this, indicating notable pairwise differences in how models rank across these two benchmarks, unearthing vulnerabilities that are less pronounced on ImageNet-C. These results demonstrate the necessity of LAION-C as a complementary benchmark for a more comprehensive evaluation of model robustness.

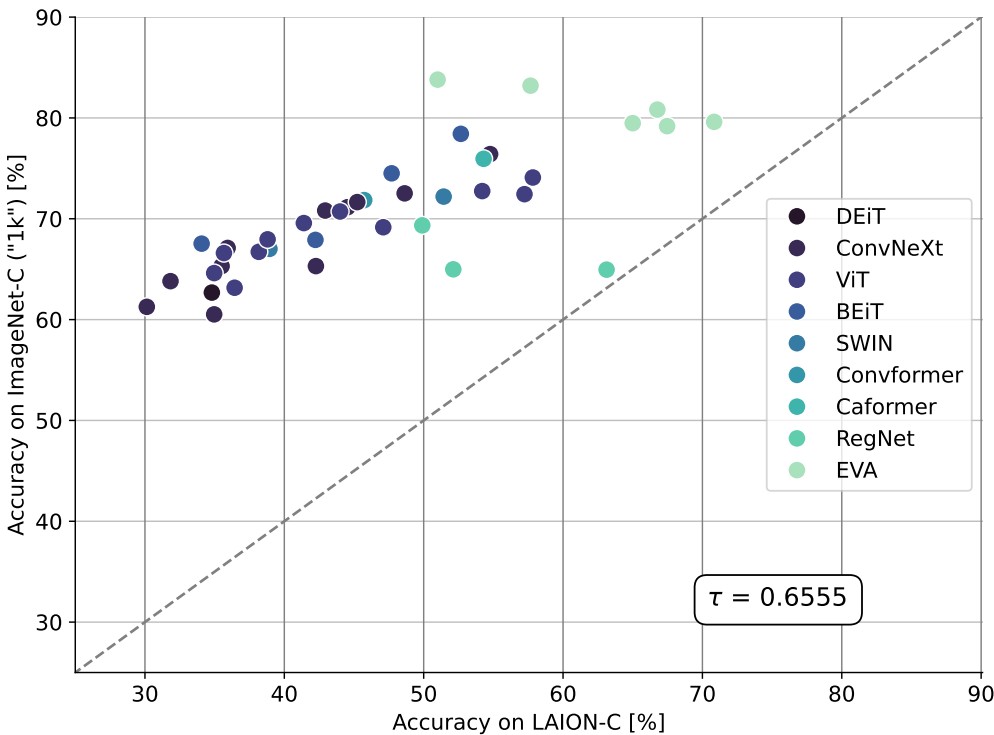

*Figure 10.* **Performance Divergence of Models on LAION-C and ImageNet-C (1k classes).** The figure illustrates the scattered performance of models across the ImageNet-C and LAION-C dataset, where a Kendall's $\tau$ coefficient of 0.66 and the shallow slope indicate a dispersed performance on LAION-C. To provide a clearer trend and to better visualize the dispersion, we supplement the suite of models with additional top-performing models sourced from the timm leaderboard (Wightman, 2024), bringing the total number of models to 40 (see Tab. 11 for a complete list).

**Occlusion and Luminance Manipulations.** For distortions involving occlusions, such as Stickers and Geometric Shapes, models usually match or exceed human performance (see second row of Fig. 11). One possible hypothesis is that this can be attributed to the robustness that models develop after e.g., masked image modeling (MIM) (Fang et al., 2023; 2024). The fact that models perform so much better than humans on partially occluded images implies that models use different features than humans. For example, for the Stickers distortion, certain ViT models outperform humans, likely due to their ability to focus on those parts of the image background that remain visible despite the occlusions. As shown in Fig. 2, the stickers occlude nearly the entire image on higher intensity levels, and little to no meaningful object information

is retained. Nevertheless, certain models are still able to correctly classify the image based on subtle background cues. This indicates that while models are performing well, they may be doing so by leveraging unintended shortcuts (Geirhos et al., 2020a), such as exploiting background information, when faced with severely occluded images. For the Luminance Checkerboard distortion, we observe that models from the ViT and EVA families outperform humans by a large margin. This advantage could potentially stem from their architectural features, such as self-attention mechanisms and patch-based processing (Fang et al., 2023; Dosovitskiy et al., 2021), which enable them to extract meaningful information from both light and dark regions independently, as well as handle subtle luminance variations. These capabilities give them a clear edge over humans and older models.

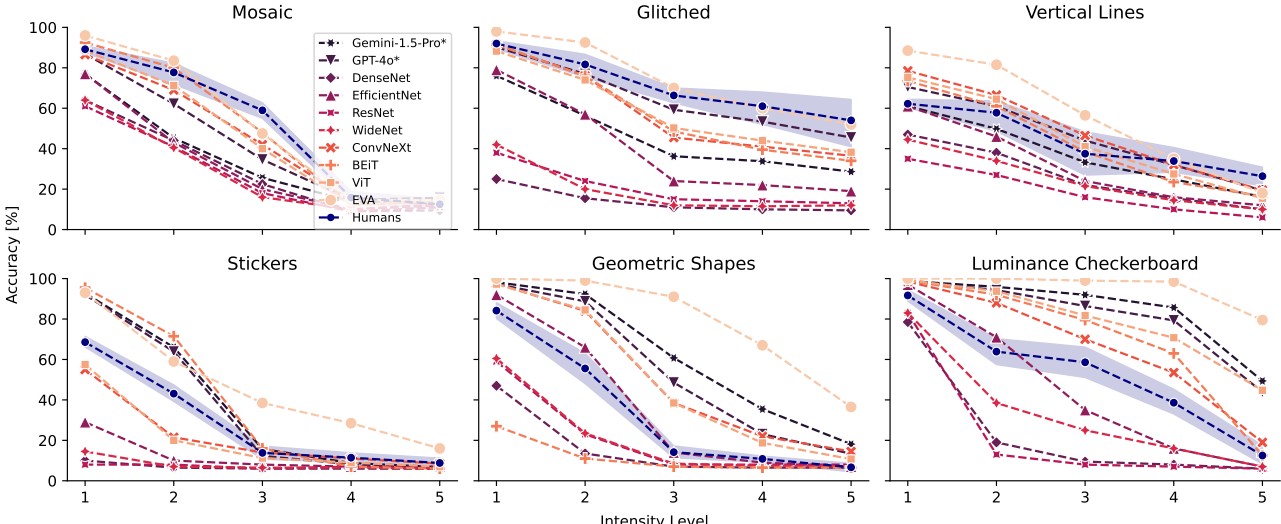

*Figure 11.* **Model performance on LAION-C.** Analogous to Fig. 6, we relate distortion intensity level to classification accuracy for the different distortions, showing the different models individually. The shaded region around human performance corresponds to the 95% confidence interval, which we omit for the models for better visibility.

*Table 10.* **LAION-C benchmark results.** Numbers show top-1 accuracy in percent. *ImageNet* refers to model accuracy on the (uncorrupted) ImageNet validation set (values sourced from the timm leaderboard (Wightman, 2024)). For each corruption, we report mean top-1 accuracy across all intensity levels, with *LAION-C* as the overall benchmark metric (averaged across corruption types). GPT-4o and Gemini 1.5 Pro were evaluated on 48,000 images, 100 for each class. For full names and descriptions, see Tab. 11 in the Appendix.

| Model | ImageNet | **LAION-C** | Mosaic | Vertical | Glitched | Luminance | Geometric | Stickers |
|---|---|---|---|---|---|---|---|---|
| EVA-G-P14-560-M30M-IN22K | 89.8 | **67.5** | 48.8 | 53.6 | 70.8 | 97.2 | **81.0** | **53.4** |
| EVA02-L-P14-448-MIM-M38M-IN22K | **90.1** | 66.8 | **53.6** | **58.2** | **78.2** | 93.6 | 76.4 | 40.6 |
| ViT-H-P14-336-CLIP-LAION-IN12K | 88.6 | 57.3 | 45.2 | 51.2 | 69.8 | 88.2 | 64.4 | 24.6 |
| ViT-L-P14-224-CLIP-OpenAI-IN12K | 88.3 | 57.8 | 52.6 | 49.8 | 68.2 | **98.6** | 55.4 | 22.4 |
| ViT-B-P32-384-CLIP-LAION-IN12K | 85.4 | 36.4 | 36.8 | 35.2 | 35.8 | 54.0 | 37.6 | 19.2 |
| ViT-B-P16-224-AugReg-IN21K | 85.5 | 47.1 | 46.4 | 42.8 | 62.0 | 71.4 | 42.4 | 17.6 |
| BEiT-v2-L-P16-224-IN1K | 87.4 | 47.7 | 52.4 | 44.8 | 63.2 | 70.2 | 11.8 | 43.8 |
| BEiT-v2-B-P16-224-IN1K | 85.6 | 42.2 | 46.2 | 40.4 | 52.6 | 68.2 | 11.4 | 34.6 |
| ConvNeXt-XXL-CLIP-LAION-IN1K | 88.6 | 54.8 | 53.0 | 53.4 | 71.8 | 77.4 | 52.2 | 20.8 |
| ConvNeXt-B-CLIP-LAION-AugReg-IN12K | 87.6 | 42.3 | 37.6 | 43.8 | 44.4 | 54.2 | 50.4 | 23.2 |
| WRN101-2-TV-IN1K | 78.8 | 21.4 | 30.4 | 28.4 | 22.0 | 22.8 | 18.2 | 6.8 |
| WRN50-2-RACM-IN1K | 81.5 | 24.0 | 26.8 | 21.4 | 17.0 | 45.0 | 24.6 | 9.4 |
| RN50-A1-IN1K | 81.2 | 19.9 | 28.0 | 18.8 | 20.8 | 23.4 | 21.2 | 7.0 |
| EFF-B3-RA2-IN1K | 82.3 | 33.2 | 32.4 | 31.8 | 40.2 | 45.2 | 37.6 | 12.2 |
| DN201-TV-IN1K | 77.3 | 19.2 | 28.6 | 26.2 | 13.2 | 23.2 | 16.8 | 7.2 |
| DN161-TV-IN1K | 77.3 | 20.0 | 31.0 | 26.8 | 15.2 | 25.2 | 15.4 | 6.6 |
| GPT-4o | - | *54.1* | *42.8* | *45.4* | *65.1* | *80.1* | *54.2* | *36.5* |
| Gemini 1.5 Pro | - | *50.2* | *34.9* | *37.0* | *46.2* | *84.4* | *60.9* | *38.1* |
| Best human observer | - | 55.2 | 58.0 | 55.3 | 78.7 | 63.4 | 40.4 | 35.7 |
| Average human observer | - | 47.0 | 50.8 | 43.6 | 71.0 | 53.1 | 34.3 | 29.1 |

## A.6. Evaluating VLMs

To evaluate GPT-4o (OpenAI, 2024) and Gemini 1.5 Pro (Team et al., 2024) on LAION-C, we decided to test a random subsample of the full dataset, consisting of 100 images per category, which were then tested on all corruptions and intensity levels, resulting in a total of 48,000 images. For ImageNet-C, we limited ourselves to only 10 images per class, to get an initial ballpark estimate of performance.

We employed the following system prompt, in line with our human experiments, during which participants were also shown examples:

---

**System Prompt:**
You are an image-recognition API.
You are always asked to classify the main object of images into one of 16 mutually exclusive categories.
In some images, the distortion may be so strong that you might not recognize anything. If you're unsure, provide your best guess - you always have to pick exactly one of the 16 categories.
The 16 categories are: primate, dog, cat, bird, fish, snake, butterfly, fruit, boat, vehicle, chair, ball, bottle, instrument, timekeeper, tool.
Here is a list of characterizations of every such category:
primate: a primate, like e.g. monkeys, chimpanzees, Orang-Utans etc.
dog: a dog, like e.g. german shepherd, labrador, golden retriever etc.
cat: a cat, like e.g. domestic cat, lion, cheetah etc.
bird: a bird, like e.g. songbird, eagle, chicken etc.
fish: a fish, like e.g. trout, shark, whale etc.
snake: a snake, like e.g. viper, cobra, seasnake etc.
butterfly: a butterfly, like e.g. monarch, cabbage butterfly, ringlet etc.
fruit: a fruit, like e.g. apple, orange, pineapple etc.
boat: a boat, like e.g. ship, gondola, fireboat etc.
vehicle: a vehicle, like e.g. truck, van, sports car etc.
chair: a chair, like e.g. bench, throne, couch etc.
ball: a ball (or a person playing with a ball), like e.g. soccer ball, football, tennis ball etc.
bottle: a bottle, like e.g. water bottle, jug, pill bottle etc.
instrument: a musical instrument (or a person playing an instrument), like e.g. sax, flute, harp etc.
timekeeper: a timekeeper, like e.g. clock, watch, sundial etc.
tool: a tool (or a person using a tool), like e.g. hammer, power drill, chainsaw etc.
Since you are an API, you always respond with minimal messages that contain exactly one word, which is the category name.

**User Prompt:**
What is the main object in this image? Categories are: primate, dog, cat, bird, fish, snake, butterfly, fruit, boat, vehicle, chair, ball, bottle, instrument, timekeeper, tool.

---

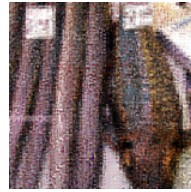

The image is highly pixelated, but I can discern the general shape of a dog, especially the head and front legs. The color pattern also suggests a dog's fur.

True label: fish
Predicted label: dog
mosaic (1)

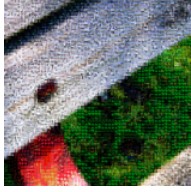

The image seems to show a fruit, possibly an apple, based on the red and green colors present in the picture.

True label: chair
Predicted label: fruit
mosaic (1)

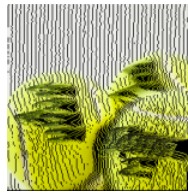

The distorted image appears to show the outline of a bird with wings outstretched. The pointed shapes resemble wings, and the overall posture suggests a bird in flight.

True label: ball
Predicted label: bird
vertical_lines (2)

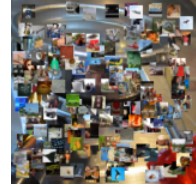

I can see pliers in the center of the image. Pliers are a tool.

True label: tools
Predicted label: tools
stickers (2)

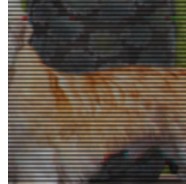

The image appears to show a bird, potentially a chicken, due to the presence of orange feathers and what looks like a wattle. The distortion makes it difficult to be certain, but the color and shape suggest a bird more than any other category.

True label: dog
Predicted label: bird
glitched (1)

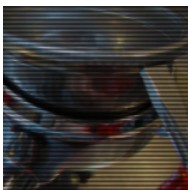

The round shape and the presence of what appears to be a drumstick suggest that this image represents a drum, which is a musical instrument.

True label: tools
Predicted label: instruments
glitched (1)

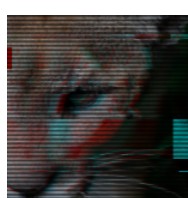

Although the image is heavily distorted, it appears to show the face of a primate. The discernible features, such as the eyes and brow ridge, suggest a primate's facial structure.

True label: cat
Predicted label: primate
glitched (2)

*Figure 12.* **Visual Reasoning in Gemini.** We provide examples of visual reasoning in Gemini-1.5-Pro, consisting of a LAION-C sample, the reasons for classification that Gemini provided and meta-information (like the final label, the ground-truth label and corruption details). In line with our findings about Error Consistency (see Fig. 8), the model is frequently confused by visual impressions that are trivial for humans to decode. Instead, it hallucinates unreasonable image properties.

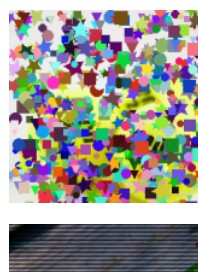

The image shows a shape that resembles the outline of a bottle, with a neck and body typical of bottles, despite the heavy distortion and colorful overlay.

True label: ball
Predicted label: bottle
geometric_shapes (3)

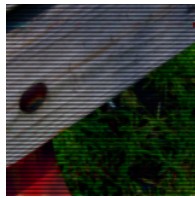

The image shows a metallic object with a handle, resembling a hammer, which is a common tool. The shape and context suggest it is used for hitting or driving nails.

True label: chair
Predicted label: tools
glitched (1)

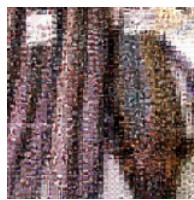

The image appears to be a mosaic or collage that forms the shape of a primate, likely a monkey or ape. The overall structure and features resemble the face and body of a primate, which is why this category is chosen.

True label: fish
Predicted label: primate
mosaic (2)

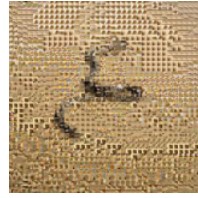

The image appears to depict a stylized or pixelated representation of a primate, likely a monkey or ape, based on the visible facial features and overall shape.

True label: snakes
Predicted label: primate
mosaic (2)

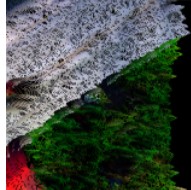

The image shows a shape and texture that resembles a fish, with a streamlined body and scales visible. The colors and patterns are typical of fish, making it the most likely category.

True label: chair
Predicted label: fish
vertical_lines (1)

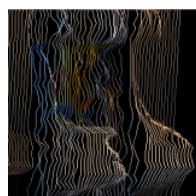

The image appears to have a distorted outline that resembles the shape of a primate, such as a monkey or ape. The contours suggest facial features and a posture typical of primates.

True label: bottle
Predicted label: primate
vertical_lines (5)

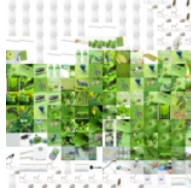

The image contains multiple small images, many of which depict butterflies. The presence of wings and colorful patterns typical of butterflies is noticeable, making 'butterfly' the most fitting category.

True label: fruit
Predicted label: butterfly
mosaic (4)

*Figure 13.* **Visual Reasoning in GPT.** Figure analogous to Fig. 12. Like Gemini, GPT has a tendency to hallucinate visual impressions. Notably, when the model makes mistakes in the sticker-corruption, it is typically led astray by the contents of the sticker-images, ignoring the background completely and failing to perceive the global structure of the image, a behavior that Gemini exhibits as well.

## A.7. Models

*Table 11.* **Model overview.** For each model used in our evaluation, we show the full model names, as used in timm, an abbreviated name used in the main text and a description of the model. While the first 16 models were used in all analyses and figures, the rest was only used for selective analyses such as Fig. 8.

| Abbreviation | Full Model Name | Description |
|---|---|---|
| EVA-G-P14-560-M30M-IN22K | eva_giant_patch14_560.m30m_ft_in22k_in1k | EVA giant model, patch size 14, pre-trained with masked image modeling (MIM) on a Merged-30M dataset, fine-tuned on ImageNet-22k and ImageNet-1k (Fang et al., 2023). |
| EVA02-L-P14-448-MIM-M38M-IN22K | eva02_large_patch14_448.mim_m38m_ft_in22k_in1k | EVA02 large model, patch size 14, pre-trained with masked image modeling (MIM) on a Merged-38M dataset, fine-tuned on ImageNet-22k and ImageNet-1k (Fang et al., 2024). |
| VIT-H-P14-336-CLIP-LAION-IN12K | vit_huge_patch14_clip_336.laion2b_ft_in12k_in1k | Vision Transformer (VIT) huge model, patch size 14, pre-trained on LAION-2B dataset using OpenCLIP, fine-tuned on ImageNet-12k and ImageNet-1k (Dosovitskiy et al., 2021). |
| VIT-L-P14-224-CLIP-OPENAI-IN12K | vit_large_patch14_clip_224.openai_ft_in12k_in1k | Vision Transformer large model, patch size 14, pre-trained on WIT-400M using CLIP, fine-tuned on ImageNet-12k and ImageNet-1k (Dosovitskiy et al., 2021). |
| VIT-B-P32-384-CLIP-LAION-IN12K | vit_base_patch32_clip_384.laion2b_ft_in12k_in1k | Vision Transformer base model, patch size 32, pretrained on LAION-2B using OpenCLIP,fine-tuned on ImageNet-12k and ImageNet-1k (Dosovitskiy et al., 2021). |
| VIT-B-P16-224-AUGREG-IN21K | vit_base_patch16_224.augreg2_in21k_ft_in1k | Vision Transformer base model, patch size 16, trained on ImageNet-21k and fine tuned on ImageNet-1k (Dosovitskiy et al., 2021). |
| BEITV2-L-P16-224-IN1K | beitv2_large_patch16_224.in1k_ft_in1k | BEiTv2 large model, patch size 16, trained on ImageNet-1k, fine-tuned on ImageNet-22k and ImageNet-1k (Bao et al., 2022; Peng et al., 2022). |
| BEITV2-B-P16-224-IN1K | beitv2_base_patch16_224.in1k_ft_in1k | BEiTv2 base model, patch size 16, trained on ImageNet-1k, fine-tuned on ImageNet-22k and ImageNet-1k (Bao et al., 2022; Peng et al., 2022). |
| CONV-XXL-CLIP-LAION-IN1K | convnext_xxlarge.clip_laion2b_soup_ft_in1k | ConvNeXt xxlarge model, pre-trained using OpenCLIP on LAION-2B, fine-tuned on ImageNet-1k (Liu et al., 2022). |
| CONV-B-CLIP-LAION-AUGREG-IN12K | convnext_base.clip_laion2b_augreg_ft_in12k_in1k_384 | ConvNeXt base model,pre-trained using OpenCLIP on LAION-2B, fine-tuned on ImageNet-12k and ImageNet-1k (Liu et al., 2022). |
| WRN101-2-TV-IN1K | wide_resnet101_2.tv_in1k | Wide ResNet-101 model, trained on ImageNet-1k, with original torchvision model weight (He et al., 2016; Zagoruyko & Komodakis, 2016). |
| WRN50-2-RACM-IN1K | wide_resnet50_2.racm_in1k | Wide ResNet-50 model, trained with RandAugment RACM recipe on ImageNet-1k (He et al., 2016; Zagoruyko & Komodakis, 2016). |
| RN50-A1-IN1K | resnet50.a1_in1k | ResNet-50 model trained on ImageNet-1k (He et al., 2016; Wightman et al., 2021). |
| EFF-B3-RA2-IN1K | efficientnet_b3.ra2_in1k | EfficientNet-B3 model, trained with RandAugment RA2 recipe on ImageNet-1k (Tan & Le, 2019). |
| DN201-TV-IN1K | densenet201.tv_in1k | DenseNet-201, DenseNet pre-trained on ImageNet-1k (Huang et al., 2017). |
| DN161-TV-IN1K | densenet161.tv_in1k | DenseNet-161, DenseNet model pre-trained on ImageNet-1k (Huang et al., 2017). |
| GPT-4o | gpt-4o-2024-08-06 | At the time of writing, the most recent snapshot of OpenAI's flagship model (OpenAI, 2024). Only evaluated on 48,000 LAION-C samples and 12,000 ImageNet-C samples. |
| Gemini-1.5-Pro | gemini-1.5-pro-002 | At the time of writing, the most recent stable version of Google's Gemini model (Team et al., 2024). Only evaluated on 48,000 LAION-C samples and 12,000 ImageNet-C samples. |
| | convnextv2_pico.fcmae_ft_in1k | |
| | convnextv2_tiny.fcmae_ft_in22k_in1k | |
| | convnext_base.fb_in22k_ft_in1k | |
| | convnext_large_mlp.clip_laion2b_augreg_ft_in1k_384 | |
| | convnext_large_mlp.clip_laion2b_soup_ft_in12k_in1k_384 | |
| | convnext_tiny.in12k_ft_in1k | |
| | convnext_small.fb_in22k_ft_in1k_384 | |
| | convnext_xlarge.fb_in22k_ft_in1k | |
| | convnext_small.in12k_ft_in1k_384 | |
| | convnextv2_large.fcmae_ft_in22k_in1k_384 | |
| | vit_betwixt_patch16_reg4_gap_256.sbb2_e200_in12k_ft_in1k | |
| | vit_mediumd_patch16_rope_reg1_gap_256.sbb_in1k | |
| | vit_wee_patch16_reg1_gap_256.sbb_in1k | |
| | vit_mediumd_patch16_reg4_gap_256.sbb2_e200_in12k_ft_in1k | |
| | vit_mediumd_patch16_reg4_gap_256.sbb_in12k | |
| | vit_pwee_patch16_reg1_gap_256.sbb_in1k | |
| | vit_betwixt_patch16_rope_reg4_gap_256.sbb_in1k | |
| | vit_betwixt_patch16_reg4_gap_256.sbb_in12k_ft_in1k | |
| | maxxvitv2_rmlp_base_rw_384.sw_in12k_ft_in1k | |
| | vgg19_bn.tv_in1k | |
| | regnety_1280.swag_lc_in1k | |
| | regnety_1280.swag_ft_in1k | |
| | regnety_320.swag_ft_in1k | |
| | inception_v3.tf_adv_in1k | |
| | beit_base_patch16_224.in22k_ft_in22k_in1k | |
| | beit_large_patch16_512.in22k_ft_in22k_in1k | |
| | deit3_large_patch16_384.fb_in22k_ft_in1k | |
| | deit_base_distilled_patch16_224.fb_in1k | |
| | swin_base_patch4_window7_224.ms_in22k_ft_in1k | |
| | swinv2_base_window12to24_192to384.ms_in22k_ft_in1k | |
| | swinv2_large_window12to24_192to384.ms_in22k_ft_in1k | |
| | eva_large_patch14_336.in22k_ft_in1k | |
| | convformer_b36.sail_in22k_ft_in1k_384 | |
| | caformer_b36.sail_in22k_ft_in1k_384 | |
| | efficientformerv2_s2.snap_dist_in1k | |

## A.8. Datasheet for LAION-C

We here include a Datasheet for LAION-C following the template proposed by (Gebru et al., 2021).

### Motivation

**For what purpose was the dataset created?** Was there a specific task in mind? Was there a specific gap that needed to be filled? Please provide a description.

The LAION-C dataset was created to serve as a benchmark for evaluating the robustness and Out-of-Distribution (OOD) generalization of large-scale vision models. It can also be used to study the difference between human and model perception.

**Who created this dataset (e.g., which team, research group) and on behalf of which entity (e.g., company, institution, organization)?**

The dataset is created by authors of this paper.

**Who funded the creation of the dataset?** If there is an associated grant, please provide the name of the grantor and the grant name and number.

Not applicable.

**Any other comments?**

None.

### Composition

**What do the instances that comprise the dataset represent (e.g., documents, photos, people, countries)?** Are there multiple types of instances (e.g., movies, users, and ratings; people and interactions between them; nodes and edges)? Please provide a description.

The instances in the LAION-C dataset represent images grouped into 16 superclasses with various synthetic distortions applied to them at 5 severity levels. Each superclass contains 273 images, and the distortions include mosaic effects, glitched images, vertical lines, geometric shapes, stickers, and luminance checkerboard patterns.

**How many instances are there in total (of each type, if appropriate)?**

In total, LAION-C consists of 131,040 images. (16 classes × 273 images × 6 corruptions × 5 severity levels.)

**Does the dataset contain all possible instances or is it a sample (not necessarily random) of instances from a larger set?** If the dataset is a sample, then what is the larger set? Is the sample representative of the larger set (e.g., geographic coverage)? If so, please describe how this representativeness was validated / verified. If it is not representative of the larger set, please describe why not (e.g., to cover a more diverse range of instances, because instances were withheld or unavailable).

The dataset is a sample of the ImageNet validation set and only contains 4,368 of the 50,000 images. As such, LAION-C is not representative of ImageNet, because it only consists of coarse superclasses. This decision was made to facilitate measuring human classification performance on LAION-C, which would not be possible with the fine-grained classes of ImageNet.

**What data does each instance consist of? "Raw" data (e.g., unprocessed text or images) or features?** In either case, please provide a description.

Each instance consists of an RGB image, as well as metadata about the ground-truth class, corruption type, and severity level, which are simply part of the filename.

**Is there a label or target associated with each instance?** If so, please provide a description.

Each image is labeled with its superclass (one of 16) and can be traced back to its original ImageNet class label.

**Is any information missing from individual instances?** If so, please provide a description, explaining why this information is missing (e.g., because it was unavailable). This does not include intentionally removed information, but might include, e.g., redacted text.

No information is missing from individual instances as each image in the dataset is synthetically altered and labeled with the type of distortion and its severity, ensuring comprehensive data for evaluation purposes.

**Are relationships between individual instances made explicit (e.g., users' movie ratings, social network links)?** If so, please describe how these relationships are made explicit.

The dataset does not contain explicit relationships between individual instances such as social links or ratings since it

primarily focuses on image recognition and distortion type evaluation without any relational context between the images.

**Are there recommended data splits (e.g., training, development / validation, testing)?** If so, please provide a description of these splits, explaining the rationale behind them.

Since the dataset is primarily used for benchmarking purposes, splitting specifics are not provided. Essentially, the entire dataset is a validation set.

**Are there any errors, sources of noise, or redundancies in the dataset?** If so, please provide a description.

The dataset is designed to introduce controlled noise through synthetic distortions to test model robustness. There are no unintentional errors or redundancies; all modifications serve the purpose of benchmark evaluation.

**Is the dataset self-contained, or does it link to or otherwise rely on external resources (e.g., websites, tweets, other datasets)?** If it links to or relies on external resources, a) are there guarantees that they will exist, and remain constant, over time; b) are there official archival versions of the complete dataset (i.e., including the external resources as they existed at the time the dataset was created); c) are there any restrictions (e.g., licenses, fees) associated with any of the external resources that might apply to a future user? Please provide descriptions of all external resources and any restrictions associated with them, as well as links or other access points, as appropriate.

The dataset is entirely self-contained.

**Does the dataset contain data that might be considered confidential (e.g., data that is protected by legal privilege or by doctor-patient confidentiality, data that includes the content of individuals non-public communications)?** If so, please provide a description.

The dataset does not contain confidential data as it is based on publicly available ImageNet data.

**Does the dataset contain data that, if viewed directly, might be offensive, insulting, threatening, or might otherwise cause anxiety?** If so, please describe why.

The dataset does not contain offensive or disturbing content as it focuses on visual distortions applied to non-sensitive images. Additionally, the images sourced from ImageNet are manually filtered to exclude any content that could be considered disturbing.

**Does the dataset relate to people?** If not, you may skip the remaining questions in this section.

Yes, the LAION-C dataset relates to people to some extent as it includes images from ImageNet, some of which feature human faces and figures. While the primary focus of the dataset is not on the individuals depicted or on analyzing human-specific data, the presence of human images means that the dataset does relate to people indirectly.

**Does the dataset identify any subpopulations (e.g., by age, gender)?** If so, please describe how these subpopulations are identified and provide a description of their respective distributions within the dataset.

The LAION-C dataset itself does not explicitly identify subpopulations by age, gender, or other demographic characteristics as part of its core design. However, since it includes images from ImageNet, which may contain human faces, there is an implicit presence of such demographic data.

**Is it possible to identify individuals (i.e., one or more natural persons), either directly or indirectly (i.e., in combination with other data) from the dataset?** If so, please describe how.

While the primary intention of the LAION-C dataset is not to facilitate the identification of individuals, it incorporates images from ImageNet, which may include human faces.

**Does the dataset contain data that might be considered sensitive in any way (e.g., data that reveals racial or ethnic origins, sexual orientations, religious beliefs, political opinions or union memberships, or locations; financial or health data; biometric or genetic data; forms of government identification, such as social security numbers; criminal history)?** If so, please provide a description.

While the LAION-C dataset primarily features synthetic distortions applied to images for technical analysis, it includes images sourced from ImageNet that may contain human faces. These images can indirectly reveal racial or ethnic origins due to the diversity of individuals depicted. However, there is no explicit focus on collecting or analyzing data related to sexual orientations, religious beliefs, political opinions, union memberships, specific locations, financial or health data, biometric or genetic data, government identification numbers, or criminal history. The in-

clusion of human images is incidental and not intended for any analysis related to these sensitive aspects.

**Any other comments?**

None.

---

## Collection Process

**How was the data associated with each instance acquired?** Was the data directly observable (e.g., raw text, movie ratings), reported by subjects (e.g., survey responses), or indirectly inferred / derived from other data (e.g., part-of-speech tags, model-based guesses for age or language)? If data was reported by subjects or indirectly inferred / derived from other data, was the data validated / verified? If so, please describe how.

The data for each instance in the LAION-C dataset is derived from ImageNet, where images are directly observable and not reported by subjects or inferred.

**What mechanisms or procedures were used to collect the data (e.g., hardware apparatus or sensor, manual human curation, software program, software API)?** How were these mechanisms or procedures validated?

First, 16 sensible high-level classes were selected that the authors deemed suitable for humans to recognize in psychophysical experiments. These classes are: ball, bird, boat, bottle, butterfly, car & truck, cat, chair, dog, fish, fruit, instrument, primate, snake, timekeeping, and tool. Then, 200 classes from the original ImageNet-1k set were selected that can constitute these high-level classes. From the pools of validation set images, 500 images were randomly selected per superclass. These images were then manually filtered to include only images that fall clearly into one of the 16 superclasses (i.e. an image showing both a ball and a dog would have been filtered out to ensure clean class labels).

**If the dataset is a sample from a larger set, what was the sampling strategy (e.g., deterministic, probabilistic with specific sampling probabilities)?**

See previous question. Candidate images from the constituent classes were sampled randomly with uniform probability.

**Who was involved in the data collection process (e.g., students, crowdworkers, contractors) and how were they compensated (e.g., how much were crowdworkers paid)?**

Details are managed by the original collector for ImageNet.

**Over what timeframe was the data collected? Does this timeframe match the creation timeframe of the data associated with the instances (e.g., recent crawl of old news articles)?** If not, please describe the timeframe in which the data associated with the instances was created.

The source dataset for the creation of LAION-C was the 2012 ILSVRC validation set ("ImageNet") which was collected over several years. The distortions applied in LAION-C were created specifically for benchmarking purposes at the time of dataset development (2023 / 2024), which do not coincide directly with the original image collection periods.

**Were any ethical review processes conducted (e.g., by an institutional review board)?** If so, please provide a description of these review processes, including the outcomes, as well as a link or other access point to any supporting documentation.

The original ImageNet dataset underwent various ethical and review processes during its development, details are managed by the original collector for ImageNet.

**Does the dataset relate to people?** If not, you may skip the remaining questions in this section.

Only indirectly. LAION-C includes images from ImageNet that feature human faces and figures.

**Did you collect the data from the individuals in question directly, or obtain it via third parties or other sources (e.g., websites)?**

Not applicable.

**Were the individuals in question notified about the data collection?** If so, please describe (or show with screenshots or other information) how notice was provided, and provide a link or other access point to, or otherwise reproduce, the exact language of the notification itself.

Not applicable.

**Did the individuals in question consent to the collection and use of their data?** If so, please describe (or show with screenshots or other information) how consent was requested and provided, and provide a link or other access point to, or otherwise re-

produce, the exact language to which the individuals consented.

Not applicable.

**If consent was obtained, were the consenting individuals provided with a mechanism to revoke their consent in the future or for certain uses?** If so, please provide a description, as well as a link or other access point to the mechanism (if appropriate).

Not applicable.

**Has an analysis of the potential impact of the dataset and its use on data subjects (e.g., a data protection impact analysis) been conducted?** If so, please provide a description of this analysis, including the outcomes, as well as a link or other access point to any supporting documentation.

No specific data protection impact analysis has been conducted for the LAION-C dataset as its primary modifications involve applying synthetic distortions like glitches to the images for technical benchmarking purposes. These alterations do not fundamentally change the nature of the data regarding privacy or ethical concerns beyond their original use in ImageNet.

**Any other comments?**

None.

---

| **Preprocessing / cleaning / labeling** |
|---|

**Was any preprocessing / cleaning / labeling of the data done (e.g., discretization or bucketing, tokenization, part-of-speech tagging, SIFT feature extraction, removal of instances, processing of missing values)?** If so, please provide a description. If not, you may skip the remainder of the questions in this section.

Images were resized to 256x256 pixels and center-cropped to 224x224 pixels, as is common for ImageNet. Images were filtered manually to ensure clean labels as described above.

**Was the "raw" data saved in addition to the preprocessed / cleaned / labeled data (e.g., to support unanticipated future uses)?** If so, please provide a link or other access point to the "raw" data.

No, LAION-C only consists of the modified images, but every filename can be uniquely traced back to the parent image from the ImageNet validation set, which

can be found here: `https://www.image-net.org/download.php`

**Is the software used to preprocess / clean / label the instances available?** If so, please provide a link or other access point.

Yes, the preprocessing, cleaning, and labeling of the dataset instances were conducted using Python. The code used for these processes is accessible via `https://github.com/FanfeiLi/LAION-C`.

**Any other comments?**

None.

---

| **Uses** |
|---|

**Has the dataset been used for any tasks already?** If so, please provide a description.

Yes, the LAION-C dataset has been utilized to evaluate the robustness and out-of-distribution (OOD) generalization capabilities of large-scale vision models.

**Is there a repository that links to any or all papers or systems that use the dataset?** If so, please provide a link or other access point.

`https://github.com/FanfeiLi/LAION-C`

**What (other) tasks could the dataset be used for?**

Beyond benchmarking vision model robustness, LAION-C could be used in studies investigating the effects of image distortions on human perception.

**Is there anything about the composition of the dataset or the way it was collected and preprocessed / cleaned / labeled that might impact future uses?** For example, is there anything that a future user might need to know to avoid uses that could result in unfair treatment of individuals or groups (e.g., stereotyping, quality of service issues) or other undesirable harms (e.g., financial harms, legal risks) If so, please provide a description. Is there anything a future user could do to mitigate these undesirable harms?

Given that the base images in the LAION-C dataset are sourced from ImageNet, which is already publicly available, the additional risk for harm is negligible.

**Are there tasks for which the dataset should not be used?** If so, please provide a description.

We would not recommend using the LAION-C dataset for fine-tuning machine learning models, due to dataset size.

**Any other comments?**

None.

---

| **Distribution** |
| :---: |

**Will the dataset be distributed to third parties outside of the entity (e.g., company, institution, organization) on behalf of which the dataset was created?** If so, please provide a description.

The LAION-C dataset will be made publicly available, allowing for distribution to third parties outside of the originating entity.

**How will the dataset will be distributed (e.g., tarball on website, API, GitHub)** Does the dataset have a digital object identifier (DOI)?

The dataset is accessible via Zenodo: `https://zenodo.org/records/14051887`

**When will the dataset be distributed?**

The dataset is already distributed.

**Will the dataset be distributed under a copyright or other intellectual property (IP) license, and / or under applicable terms of use (ToU)?** If so, please describe this license and / or ToU, and provide a link or other access point to, or otherwise reproduce, any relevant licensing terms or ToU, as well as any fees associated with these restrictions.

LAION-C will be available under a CC BY-NC 4.0 license, allowing non-commercial use with proper attribution only, to ensure compliance with the original ImageNet license.

**Have any third parties imposed IP-based or other restrictions on the data associated with the instances?** If so, please describe these restrictions, and provide a link or other access point to, or otherwise reproduce, any relevant licensing terms, as well as any fees associated with these restrictions.

The original ImageNet data is subject to terms of access that limit its use to non-commercial research and educational purposes only. The full terms of access can be found here: `https://www.image-net.org/download.php`

**Do any export controls or other regulatory restrictions apply to the dataset or to individual instances?** If so, please describe these restrictions, and provide a link or other access point to, or otherwise reproduce, any supporting documentation.

Since the images are modified ImageNet images, the restrictions of the ImageNet license apply.

**Any other comments?**

None

---

| **Maintenance** |
| :---: |

**Who will be supporting / hosting / maintaining the dataset?**

The dataset is maintained by the authors of the associated paper.

**How can the owner / curator / manager of the dataset be contacted (e.g., email address)?**

For questions regarding the dataset, please contact Fanfei Li at `<first.last@tuebingen.mpg.de>`.

**Is there an erratum?** If so, please provide a link or other access point.

There is not an explicit erratum as for now.

**Will the dataset be updated (e.g., to correct labeling errors, add new instances, delete instances)?** If so, please describe how often, by whom, and how updates will be communicated to users (e.g., mailing list, GitHub)?

Information will be provided upon publication.

**If the dataset relates to people, are there applicable limits on the retention of the data associated with the instances (e.g., were individuals in question told that their data would be retained for a fixed period of time and then deleted)?** If so, please describe these limits and explain how they will be enforced.

Not applicable (beyond agreements made for ImageNet).

**Will older versions of the dataset continue to be supported / hosted / maintained?** If so, please describe how. If not, please describe how its obsolescence will be communicated to users.

Should newer versions of the dataset be created, older versions will continue to be available via Zenodo.

**If others want to extend / augment / build on / contribute to the dataset, is there a mechanism for**

**them to do so?** If so, please provide a description. Will these contributions be validated / verified? If so, please describe how. If not, why not? Is there a process for communicating / distributing these contributions to other users? If so, please provide a description.

We encourage other researchers to build on LAION-C, for example by contributing their own corruptions. While there is no automatic mechanism (such as publicly accessible version control, e.g. via Github) for this, we encourage interested parties to reach out to the authors.

**Any other comments?**

None.

