# OpenReview forum: "LAION-C: An Out-of-Distribution Benchmark for Web-Scale Vision Models"
_ICML.cc/2025/Conference — ICML 2025 poster_

### Official Review · Reviewer_ZcBa · 2025-02-27

**Overall Recommendation:** 4

**Summary:**

This paper presents a novel dataset, designated as LAION-C, where the letter C denotes "corrupted." This dataset bears similarities to ImageNet and ImageNet-C.
The dataset under consideration contains six corruptions (Mosaic, Glitched, Vertical Lines, Stickers, Geometric Shapes, Luminance Checkerboard) and 16 superclasses.
Rather than focusing on natural corruptions, the objective is to create a highly synthetic corrupted dataset.
The evaluation process entails the psychophysical observation of 19 participants.

## update after rebuttal
None

**Claims And Evidence:**

The correlation between model size and robustness is positive; that is to say, the larger the model, the more robust it becomes.
This assertion is supported by empirical evidence from experiments conducted on vision foundation models, which involved the utilization of 19 human evaluators. However, it is noteworthy that the performance of these human evaluators frequently surpasses that of the models.

**Essential References Not Discussed:**

None.

**Ethical Review Flag:**

Flag this paper for an ethics review.

**Experimental Designs Or Analyses:**

The primary concern pertains to the composition of the evaluator cohort, which consists of 19 individuals.
A comparison between ImageNet-C and LAION-C demonstrated the significance of this factor.

**Methods And Evaluation Criteria:**

The provided dataset is a benchmark for out-of-distribution (OOD) detection. It has been observed that the accuracy of the models decreases to a greater extent than in ImageNet-C; therefore, the utilization of this dataset is logical.

**Other Comments Or Suggestions:**

None

**Other Strengths And Weaknesses:**

Strengths
- This dataset can be a new challenge in OOD detection.
- Model often outperform humans.
- Their results show that vision foundation models evaluated on ImageNet-C are robust.
But evaluated on LION-C they show less accuracy.
- Dataset was manually filtered. To ensure that only one object is on the image.
- Mechanical writing.

Weaknesses
- Only 16 classes: Some OOD detectors could be class dependant, and fewer classes could be easier for them.
- The explanation why artificial corruption are important.
- there is no categorization of distribution shift. Covariate? [1]

[1] https://arxiv.org/pdf/2501.18463v2

**Questions For Authors:**

- Did you take into account that the ImageNet has .png (validation set) images and the Laion-C has .jpg compressed images?
- Are 19 evaluators a good number?
- Why only 16 classes? Isn't that easier for some OOD detectors? The hierarchical 1000 classes of ImageNet would be more difficult if the detector takes the class probabilities into account?
- There are different intensity levels. The ImageNet-C does not have this. If I want to use this dataset, how should I use the different intensity levels?

**Relation To Broader Scientific Literature:**

OOD detectors have become better and is still a challenge.

**Theoretical Claims:**

None

---

> ### Author Rebuttal · Authors · 2025-03-31
>
> Dear Reviewer,
>
> Thank you for your favourable review. We appreciate that you find the utilization of LAION-C **logical** because it **poses a new challenge for OOD-evaluation**, and **value the human experiment**. Prompted by your feedback, we will update the manuscript to include a section motivating our artificial corruptions better, and publish a version of our benchmark that consists of 1,000 class labels (LAION-C-1K). Thank you for alerting us to the existing nomenclature regarding the different types of distribution shifts (semantic vs covariate shifts); we have updated the manuscript accordingly. Our shift is indeed of the covariate shift kind.
>
>
> *Q: “Did you take into account that the ImageNet has .png (validation set) images and the Laion-C has .jpg compressed images?”*
>
> A: Thank you for raising this point. We double-checked that ImageNet images are JPEG-compressed in both training and validation sets. Since LAION-C base images were taken from the ImageNet validation set, analogous to ImageNet-C, this design choice matches other benchmarks. We therefore also use JPEG-compression, setting parameters which give close to lossless compression.
>
> *Q: “Are 19 evaluators a good number?”*
>
> A: Valid question. Yes, 19 evaluators are a sufficient and typical number for this type of study: Since each human observer is exposed to 2 corruptions, every corruption is seen by at least 6 different humans, assuring good coverage. For reference, [1] used only 4 humans per corruption for most of their corruptions. In psychophysics, this type of design is called a “small N design” (fewer participants that see many images, [3]). More quantitatively, we see in Fig. 10 that the 95% confidence intervals surrounding our estimate of the average human performance are quite small (+- 4.61%). Statistically speaking, it is therefore very unlikely that adding more human observers would change any of the  results or interpretations. We will add a detailed discussion of this to the Appendix. Since beating the best human is more difficult than beating the average human, we analyze peak performance rather than average performance in Fig. 5 and it is impressive to see that good models beat every single human. The confidence interval is smaller for the models because they were evaluated on all images, while each human only saw a subset of images for practical reasons. In the Appendix, we will now include a comparison to both average and peak human performance."
>
> *Q: “Why only 16 classes?”*
>
> A: We used 16 classes to enable a comparison to human observers in line with earlier work [1, 2], because humans cannot be expected to reliably classify images into too many categories, simply due to pragmatic limitations (size of the response icons, time per trial etc.) Prompted by your suggestions, we have decided to also provide a version of our dataset consisting of all 1,000 ImageNet classes, which we call LAION-C-1k. The dataset is ready and has been submitted for review on Zenodo. While we are not including a direct link at this stage respecting the rebuttal policies, we will link the dataset in the camera-ready version.
>
> *Q: “How should I use the different intensity levels?”*
>
> A: We included different intensity levels so that practitioners would be able to see the decline in model performance and obtain a more fine-grained evaluation of robustness. This is done by some other OOD datasets as well, like model-vs-human [1] and ImageNet-C. The protocol is to evaluate a new model on all levels of our benchmark and report the average across levels for the benchmark result (we’ll make sure to make this clear in the manuscript and github to ensure comparability). Additionally, users are free to report performance curves like we do in e.g. Fig. 5.
>
> [1]: Partial success in closing the gap between human and machine vision (Geirhos et al., 2021)
>
> [2]: Generalisation in humans and deep neural networks (Geirhos et al., 2018)
>
> [3] Small is beautiful: In defense of the small-N design (Smith and Little, 2018)

---

> > ### Comment · Reviewer_ZcBa · 2025-04-04
> >
> > Q: “Did you take into account that the ImageNet has .png (validation set) images and the Laion-C has .jpg compressed images?”
> > Well, but if you add the corruptions to the image and safe again as jpeg. Shouldn't it compressed again?
> >
> > Thanks for clarifying the other questions. It is more understandable now.

---

> > > ### Author Response · Authors · 2025-04-07
> > >
> > > Thank you for your response. We are pleased that our previous responses have helped clarify your questions. We export the corrupted images with parameters subsampling=0 and quality=100 using PIL. These parameters should result in an effectively lossless compression, making the effect of compression negligible. Saving images in JPEG format significantly saves on storage space and makes the data more accessible, and is therefore  standard practice. Since ImageNet itself consists of JPEG images, there should not be detrimental effects due to this format.

---

### Official Review · Reviewer_xiHt · 2025-03-02

**Overall Recommendation:** 2

**Summary:**

This paper proposed a new dataset called LAION-C for evaluating image classification model robustness against out-of-distribution data. The LAION-C datasets select samples from ImageNet validation set, and added with 6 different types of synthetic distortions, each with 5 different levels of severity. A substantial evaluation was conducted to assess different commonly used neural network models performance on LAION-C dataset. A human review was also conducted to compare human and model performance.

**Claims And Evidence:**

1. One claim from the paper is that the state-of-the-art image recognition models were able to perform well on OOD datasets, while degrades substantially on LAION-C, which suggests LAION-C being a valuable dataset to test model performance for OOD datasets. This is supported by results in Figure 3. There is a consistent performance degradation for all the evaluated models from ImageNet-C to LAION-C. I think purely from performance point of view, LAION-C is a more challenging benchmark than ImageNet-C.
2. Another claim from the paper is that models fine-tuned on LAION-C dataset can improve the performance that is comparable to clean dataset performance. This is supported by results in Table 1. This looks promising except for category Stickers and Mosaic Shape. It would be helpful to provide more insights on why these two types of distortions remain more challenging than other distortions. To be more specific, Stickers and Geometric Shape distortions both block information from clean image, but Geometric Shape distortion can be handled very well with re-training. Similarly, Mosaic and Vertical Lines obfuscate information, but Vertical Lines performance improves significantly with retraining.
3. Another claim from the paper is that models can achieve comparable or superior performance in some OOD classes than human, while still struggle on other categories. For this claim, the evidence was provided in Figure 5. For this claim, more justification on the human performance is needed. We can see the best model performance has a much narrower 95% confidence interval compared to human. Is the best human performance based on aggregation of all the participates, or is it based on overall best individual performer? Given the small subject population and no-expertise required for participating the analysis, it would be more proper to consider claiming on average human performance rather than best human performance.
4. Another claim from the paper is that LAION-C dataset is a challenging dataset but can potentially be solved. This is also supported by Table 1. I have some doubts on this claim as we cannot tell if the improved performance of model trained with LAION-C poses any regression on the clean data performance. In other words, it is unclear whether the model overfits to the distortion introduced in LAION-C or not. It would be helpful to show the same model (trained with LAION-C data) performance on clean and distorted images side-by-side.

**Essential References Not Discussed:**

Some of the references that can be added are

[1] Miyai, Atsuyuki, et al. "Locoop: Few-shot out-of-distribution detection via prompt learning." Advances in Neural Information Processing Systems 36 (2023): 76298-76310.
[2] Yang, Jingkang, et al. "Generalized out-of-distribution detection: A survey." International Journal of Computer Vision 132.12 (2024): 5635-5662.

**Experimental Designs Or Analyses:**

Centering around the proposed LAION-C dataset, the paper designed a series of experiments to answer following questions: 1) How challenging is the LAION-C dataset to modern image recognition models? 2) How human performs on LAION-C dataset? 3) Is there any difference in errors made by model and human? 4) Can LAION-C dataset be solved by recognition model? I found the experimental design is in general thoughtful and comprehensive. For example, on evaluating the extend of OOD of LAION-C, the authors utilized three different approaches including qualitative assessment, quantitative measurement of model performance, and quantitative measurement of difference between datasets.

The main concern I have though is all the analysis and discussion anchors on difference from ImageNet-C. This is probably originated from the fact that LAION-C use original images from ImageNet. There is not much discussion or contrast with other OOD dataset. Furthermore, model OOD work often use cross-dataset evaluation for evaluating model capability on handling OOD data. For example, in [1] where the model is trained on ImageNet-1K and tested on a bunch of other datasets.

[1] Miyai, Atsuyuki, et al. "Locoop: Few-shot out-of-distribution detection via prompt learning." Advances in Neural Information Processing Systems 36 (2023): 76298-76310.

**Methods And Evaluation Criteria:**

1. The distortions were manually designed and programmatically generated. While it does pose challenge for current recognition models, I'm a bit skeptical about the practicality of these kind of distortion in real-world use case. It would be good to give motivation on why choosing these 6 types of distortions.
2. The LAION-C uses clean images from ImageNet. If so, why it is called LAION-C? Shouldn't it be called another variant of ImageNet? More importantly, LAION dataset is known for its web-scale and diversity. But LAION-C is neither from web-scale images or diverse in terms of image classes. I think calling it LAION-C is misleading.
3. The LAION-C dataset only have 273 clean original images. I understand that to have manageable human review experiment, we may not be able to use large scale dataset. But for a final release, I would suggest consider extending the number of images and number of classes further.

**Other Comments Or Suggestions:**

Typo in supplementary materials: 131.040 should be 131,040.

**Other Strengths And Weaknesses:**

[Other strengths]
+ A variety of model image recognition models were evaluated and compared for performance on LAION-C
+ The error consistency analysis between human and models is insightful to show different failure modes of human vision and computer vision.
+ The appendix provides a comprehensive discussion on what is LAION-C and how it is constructed using the template format.

[Other weaknesses]
- There is no justification from practical or theoretical point of view why using the 6 proposed distortions. The proposal of distortion types seem a bit arbitrary. It is unclear about the implication of a model performing well on LAION-C dataset.
- There is limited novelty in terms of dataset construction in the three aspects: 1) the clean images are selected from existing ImageNet dataset; 2) the way of constructing OOD samples is based on existing practice by artificially adding obstruction to or alternating the image, which has been done in ImageNet-C. The only difference is the type of distortion; 3) there is no new proposals on the evaluation of OOD performance, which is still based on multi-class classification accuracy..

**Questions For Authors:**

1. I do not fully understand why this proposed dataset is called LAION-C while all its original clean images come from ImageNet. In addition, the number of original images is only 273 from 16 classes, even though there are a total of 131,040 images accounting for all distortion variations. So I don't think it resembles the diverse collection and scale of LAION dataset. It would be helpful if the authors can clarify the rationale of naming the dataset as LAION-C instead of another variant of ImageNet. It seems the same kind of distortion can be applied to LAION images as well, in which case the resulting dataset can also be called 'LAION-C'. Why not?
2. On the justification of LAION-C can be solved, I don't think the Table 1 result is sufficient. A critical question is whether the improvement of LAION-C accuracy can be achieved without regression on clean dataset. It would be good to evaluate the same model performance on clean dataset (see more discussion in 'Claims And Evidence'). If there is no regression on clean data, that means model performance on LAION-C can simply be improved by adding distortion as an augmentation. It is indeed promising to solve LAION-C. Otherwise, the challenge of achieving robustness on unseen distortions remain unsolved and more evidence is needed on whether LAION-C can be solved.
3. OOD evaluation in recent literatures are typically based on using distinct training and testing datasets such as training under ImageNet and testing on iNaturalist. There is no discussion or comparison of LAION-C with this popular OOD evaluation protocol. It may be that model performance on ImageNet-C becomes more saturated, but I'm not sure it is the case that such cross-dataset evaluation also becomes saturated. In that sense, the claim on modern image recognition models saturated on current benchmark is not well supported. Could the authors clarify what is the level of difficulty of evaluating on LAION-C compared to other cross-dataset evaluation protocol?
4. Related to point 3, what is the main novelty of constructing LAION-C?
5. Related to point 1 and 2, what is the significance of solving LAION-C? Can we draw a better conclusion that if a model does a better job on LAION-C, it has better robustness against OOD data regardless of its performance on other OOD evaluation benchmark? If not, what other insights we can gain from the performance on LAION-C dataset?

**Relation To Broader Scientific Literature:**

When discussing related work, the focus of this paper in on contrasting the LAION-C dataset and ImageNet based dataset. There is no discussion on other OOD datasets. In addition, there is no experiment design related to cross-dataset evaluation protocol, which is often adopted in OOD literature.

**Theoretical Claims:**

There is no theoretical claims.

---

> ### Author Rebuttal · Authors · 2025-03-31
>
> Dear Reviewer,
>
> thank you for taking the time to provide such detailed and insightful feedback. We were delighted to read that you found our **experimental design [to be] in general thoughtful and comprehensive**, and appreciated the **substantial evaluation and insightful error analysis**. We have clarified the manuscript based on your suggestions, added citations, and scaled the dataset to a substantially larger version with 1k classes, for a total of 1.5M images.
>
> *Q: Motivation for choosing these 6 types of distortions.*
>
> Our distortions needed to fulfill the desiderata of 1. being exotic enough to have a low occurrence probability even in web-scale datasets and 2. testing relevant feature extraction capabilities of the models. We now include this motivation in Sec 2.1.
> To solve the Stickers and Mosaic distortion, a model needs to be able to holistically integrate the image, instead of being led astray by local image cues induced by sub-images, which is notoriously difficult for DNNs [1].
> The Glitch and Vertical Lines distortions are the most exotic and globally disruptive image transformations we could find, and destroy the texture cues that models rely on [1].
> The Geometric Shapes distortion tests amodal completion, a staple of human visual processing even in infants [2,3]. They also change the color distribution of the image, which humans are robust to because we do not rely primarily on color for object recognition [4,5].
> The Luminance Checkerboard distortion tests a model’s ability to adapt to local lighting conditions, an important skill of the human visual system [6].
>
> [1] Geirhos et al 2019
>
> [2] Kellmann and Spelke 1983
>
> [3] Nanay 2018
>
> [4] Tanaka and Presnell 1999
>
> [5] Biedermann and Ju 1988
>
> [6] Heeger 1992
>
> *Q: not much discussion or contrast with other OOD dataset*
>
> Thanks for the suggestion - we have added a figure contrasting LAION-C with other OOD datasets: https://ibb.co/HfZmLkTd. This shows that LAION-C offers better resolution of model differences. Furthermore, in the camera-ready version we will substantially expand our discussion of other OOD datasets.
>
> *Q: I don't think the Table 1 result is sufficient.*
>
> Thanks for the great suggestion. While our initial approach demonstrates learnable signals in LAION-C, your suggestion offers a stronger proof of concept. We now fine-tuned the same model backbone as in Tab.1 on a mixture of clean and distorted ImageNet-Train images and evaluated on both ImageNet-Val 16 class and LAION-C 16 class. The results illustrated here (https://ibb.co/rGmQk2rG) indeed show that *‘improvement of LAION-C accuracy can be achieved without regression on the clean dataset’*. We will update Sec. 3.3 accordingly with the results and full experiment setup to strengthen our argument.
>
> *Q: LAION-C only has 273 images*
>
> We selected 273 images *per class*, yielding 4,368 base images and ~130k total images after corruption at different intensity levels. Nonetheless, we hear your point regarding broad coverage and scale. To address it, we extended our dataset to the full 50k IN-validation set, resulting in LAION-C-1k with 1.5 million images. The dataset is ready and has been submitted for review on Zenodo. While we are not including a direct link at this stage respecting the rebuttal policies, we will link the dataset in the camera-ready version.
>
> *Q: Why is it called LAION-C?*
>
> Great question. We follow common practice in the OOD literature, where datasets are named w.r.t. the dataset for which they’re OOD. E.g., ImageNet-Sketch is designed to be OOD for ImageNet (even though none of the sketch images are part of ImageNet since that would defeat the purpose of being OOD). We now explain this motivation in the introduction. Applying corruption to LAION images directly is possible, but they don’t have class labels, thus cannot be used (without proxy/pseudo labels) for classification.
>
> *Q: [M]ore justification on the human performance is needed.*
>
> Please see our response to ZcBa.
>
> *Q: LAION-C difficulty vs other protocols?*
>
> Our main results (Fig. 4) follow a cross-dataset evaluation protocol: The models were trained on web-scale datasets, fine-tuned on IN-1k but evaluated on LAION-C. In general, it is possible to apply any OOD evaluation protocol to LAION-C that one could also apply to other OOD-datasets, such as ImageNet-[A, R, C, Sketch].
>
> *Q: Why are sticker and mosaic distortions more challenging?*
>
> Great question. We introduced content-rich image tiles into the original image, to test the model's ability to process images holistically in spite of local distractions. These two distortions are built on simpler augmentations that mask or pixelate content, as illustrated in https://ibb.co/CpBvR2Y3. We observe an up to 35% drop on ViT-B32 compared to these easier distortions (https://ibb.co/hJj8pfQw), indicating that local signals pose a significant challenge. Fig.12 also supports this, as LLM-based models frequently direct attention toward the inserted tiles.

---

### Official Review · Reviewer_qKNV · 2025-03-18

**Overall Recommendation:** 3

**Summary:**

This paper introduces LAION-C, a new benchmark dataset for evaluating out-of-distribution (OOD) robustness of web-scale vision models. The authors argue that existing benchmarks like ImageNet-C are no longer sufficiently OOD for models trained on massive web datasets like LAION, as these models are likely exposed to similar corruptions during training. LAION-C proposes six novel, synthetic distortion types designed to be genuinely OOD even for web-scale models. The paper presents evaluations of various state-of-the-art models, including large language models with vision capabilities, on LAION-C and compares model performance to human psychophysical data, suggesting a paradigm shift where models are now matching or exceeding human OOD robustness in certain scenarios.

**Claims And Evidence:**

The authors report that existing OOD evaluation benchmarks such as ImageNet-C are not truly OOD anymore as LAION-based datasets basically train on whole web. This fact is anecdotally supported in Figure 1, performance of zero-shot models is compared between the ImageNet-C test set and the LAION-C one to show LAION-C is harder to solve and finally, a FID score is computed to support representation differences between datasets.

These claims adequately support the limitations of ImageNet-C and the added difficulty of LAION-C for zero-shot models.

**Essential References Not Discussed:**

[2] Hendrycks, Dan, et al. "Natural adversarial examples." Proceedings of the IEEE/CVF conference on computer vision and pattern recognition. 2021.

[3] Wang, Haohan, et al. "Learning robust global representations by penalizing local predictive power." Advances in neural information processing systems 2019.

**Experimental Designs Or Analyses:**

Good design, other benchmarks OOD than ImageNet-C should be added and compared with.

**Methods And Evaluation Criteria:**

The authors evaluate a large suite of vision models as well as some experiments with human discriminators to put in parallel the difficulty increase between human and machine when increasing the perturbation intensity.

More ImageNet OOD variants could have been studied such as  ImageNet-Adversarial [1] or ImageNet-Sketch [2] which would represent more naturally occurring OOD perturbations.

[2] Hendrycks, Dan, et al. "Natural adversarial examples." Proceedings of the IEEE/CVF conference on computer vision and pattern recognition. 2021.

[3] Wang, Haohan, et al. "Learning robust global representations by penalizing local predictive power." Advances in neural information processing systems 2019.

**Other Comments Or Suggestions:**

The paper oversells the "paradigm shift" claim as the perturbations proposed and in the same vein as the ImageNet-C corruptions.

**Other Strengths And Weaknesses:**

The artificial perturbations proposed are effective at degrading the performance of zero-shot models, loosely align to human perception (Figure 5) and have been shown to be fair because learnable (Table 1).

One limitation though is that the synthetic nature of the perturbations questions whether they represent real world occurrences to test actual failure cases of zero-shot models when deployed to the real world.

In this scope, a complementary approach similar to ImageNet-Adversarial but applied to LAION would be relevant.

**Questions For Authors:**

Have the authors considered constructing a OOD test set in a similar manner to the ImageNet-A dataset but for LAION to test real world failure cases ?

**Relation To Broader Scientific Literature:**

This research is relevant to the community as it shows the limits of the ImageNet-C benchmark for OOD robustness and proposes an alternative.

**Theoretical Claims:**

No theoretical claims

---

> ### Author Rebuttal · Authors · 2025-03-31
>
> Dear Reviewer,
>
> thank you for your insightful comments! We are delighted that you find our work to have a **good experimental design** and consider it **relevant to the community**.
>
>
> *Q: “More ImageNet OOD variants could have been studied / compared with”*
>
> A: Thanks for this excellent suggestion. We have added a comparison of LAION-C and several well-established OOD datasets to the paper, see https://ibb.co/HfZmLkTd, which shows that LAION-C captures the variance in model performance better than other datasets, with a standard deviation of ~27%, whereas other common OOD datasets, on average, have only ~10%. LAION-C is tested on a 16-class basis, while other datasets typically use 200-1000 classes, making this result even more remarkable. We want to note, however, that the point of LAION-C is precisely not to capture “naturally occurring” perturbations (see next question). We also added the missing reference to ImageNet-Sketch you pointed out (note the related work section in Appendix A.1, where we already refer to ImageNet-A).
>
>
> *Q: “[T]he synthetic nature of the perturbations questions whether they represent real world occurrences [...]”*
>
> A: Indeed, LAION-C is designed to be synthetic, just like ImageNet-C for ImageNet. “Natural” images as used by other common OOD datasets are scraped from the web, making them no longer OOD in an era where large multimodal foundation models are trained on nearly the entire web, for multiple epochs: as of 2025, we need to assume that most natural images found on the web were part of the training corpus. Robust representations, however, should generalize to challenging images (e.g. identity-preserving transformations) no matter the type, both synthetic and natural—just like human perception is robust to many different types of OOD data. Being able to handle challenging synthetic input typically transfers to unexpected natural input, as evidenced by the “sim2real” line of work e.g. in robotics and autonomous driving.
>
>
> *Q: “Have the authors considered constructing a OOD test set in a similar manner to the ImageNet-A dataset but for LAION to test real world failure cases?”*
>
> A: Thank you for the suggestion. In the current era of large-scale training, we have to assume that models will eventually be trained on the entire internet, hence scraping the web for benchmark images, as was done for ImageNet-A, is no longer a feasible choice for obtaining an OOD dataset.
>
>
> *Q: “The paper oversells the paradigm shift claim as the perturbations proposed are in the same vein as the ImageNet-C corruptions.”*
>
> A: This might be a misunderstanding. The “paradigm shift” we refer to is the shift from humans outperforming models (as observed in prior work [1,2,3,4,5]) whereas now, based on our carefully collected human comparison data for LAION-C, the best models outperform humans. Apologies if that wasn’t clear, we’ll make sure to highlight and explain this better.
>
> [1] Partial success in closing the gap between human and machine vision (Geirhos et al., 2021)
>
> [2] Comparing deep neural networks against humans: object recognition when the signal gets weaker (Geirhos et al., 2017)
>
> [3] ViewFool: Evaluating the Robustness of Visual Recognition to Adversarial Viewpoints (Dong et al., 2023)
>
> [4] Recognition in Terra Incognita (Beery et al., 2018)
>
> [5] Why do deep convolutional networks generalize so poorly to small image transformations? (Azulay & Weiss, 2019)

---

### Decision · Program_Chairs · 2025-05-01

**Decision:**

Accept (poster)

**Comment:**

This paper received two positive reviews, and one weak reject where the reviewer, xiHt, acknowledged that the rebuttal addressed most of his concerns. The AC agrees that the rebuttals were convincing, and the paper should be accepted. It introduces a large, new dataset for OOD detection that modernizes the field beyond the typical ImageNet benchmarks by addressing scaling limitations. The human study of OOD detection is also a useful contribution.